# The Geometry of Neural Nets' Parameter Spaces Under Reparametrization

**Agustinus Kristiadi**      **Felix Dangel**
Vector Institute, University of Tübingen
akristiadi,fdangel@vectorinstitute.ai

**Philipp Hennig**
University of Tübingen, Tübingen AI Center
philipp.hennig@uni-tuebingen.de

## Abstract

Model reparametrization, which follows the change-of-variable rule of calculus, is a popular way to improve the training of neural nets. But it can also be problematic since it can induce inconsistencies in, e.g., Hessian-based flatness measures, optimization trajectories, and modes of probability densities. This complicates downstream analyses: e.g. one cannot definitively relate flatness with generalization since arbitrary reparametrization changes their relationship. In this work, we study the invariance of neural nets under reparametrization from the perspective of Riemannian geometry. From this point of view, invariance is an inherent property of any neural net *if* one explicitly represents the metric and uses the correct associated transformation rules. This is important since although the metric is always present, it is often implicitly assumed as identity, and thus dropped from the notation, then lost under reparametrization. We discuss implications for measuring the flatness of minima, optimization, and for probability-density maximization. Finally, we explore some interesting directions where invariance is useful.

## 1 Introduction

Neural networks (NNs) are parametrized functions. Since it is often desirable to assign meaning or interpretability to the parameters (weights) of a network, it is interesting to ask whether certain transformations of the parameters leave the network *invariant*—equivalent in some sense. Various notions of invariance have been studied in NNs, in particular under weight-space symmetry [5, 14, 20, 42, 53] and reparametrization [20, 21, 30, 32, 71]. The former studies the behavior of a function $\mathcal{L}(\boldsymbol{\theta})$ under some invertible $T : \Theta \times \mathcal{G} \to \Theta$ where $\mathcal{G}$ is a group; for any $\boldsymbol{\theta}$, the function $\mathcal{L}$ is *invariant under the symmetry $T$* if and only if $\mathcal{L}(\boldsymbol{\theta}) = \mathcal{L}(T(\boldsymbol{\theta}, g))$ for all $g \in \mathcal{G}$. For example, normalized NNs are symmetric under scaling of the weights, i.e. $\mathcal{L}(\boldsymbol{\theta}) = \mathcal{L}(c\boldsymbol{\theta})$ for all $c > 0$ [7]—similar scale-symmetry also presents in ReLU NNs [61]. Meanwhile, *invariance under reparametrization* studies the behavior of the NN when it is transformed under the change-of-variable rule

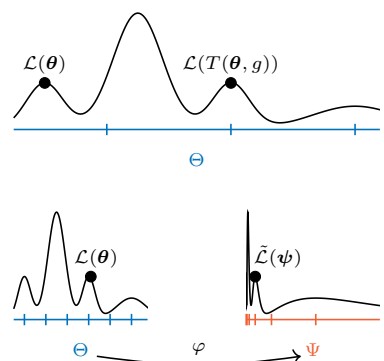

Figure 1: The difference between the symmetry (**top**) and reparametrization problems (**bottom**). $\Theta$ and $\Psi$ are two different parameter spaces.

of calculus: Given a transformed parameter $\boldsymbol{\psi} = \varphi(\boldsymbol{\theta})$ under a bijective differentiable $\varphi : \Theta \to \Psi$ that maps the original parameter space onto another parameter space, the function $\mathcal{L}(\boldsymbol{\theta})$ becomes $\hat{\mathcal{L}}(\boldsymbol{\psi}) := \mathcal{L}(\varphi^{-1}(\boldsymbol{\psi}))$. Note their difference (see Fig. 1): In the former, one works on a single parameter space $\Theta$ and a single function $\mathcal{L}$—the map $T$ acts as a symmetry of elements of $\Theta$ under the group $\mathcal{G}$. In contrast, the latter assumes two parameter spaces $\Theta$ and $\Psi$ which are connected by $\varphi$ and hence two functions $\mathcal{L}$ and $\hat{\mathcal{L}}$.

37th Conference on Neural Information Processing Systems (NeurIPS 2023).

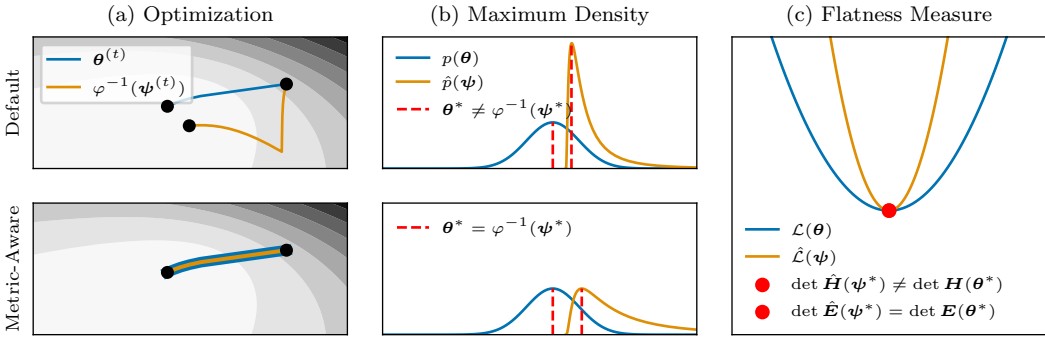

Figure 2: Invariance is retained (**bottom**) if the Riemannian metric is explicitly tracked and one transforms geometric objects such as vectors, covectors, and the metric itself properly under a reparametrization $\varphi : \boldsymbol{\theta} \mapsto \boldsymbol{\psi}$. **(a)** Gradient descent's trajectories are therefore invariant under a change of parametrization. **(b)** Invariance of the modes of a probability density function is an inherent property under a natural base measure induced by the metric. **(c)** When the Hessian $\boldsymbol{H}$ at a critical point is seen as a linear map $\boldsymbol{E}$ with the help of the metric, its determinant is invariant.

We focus on the latter. While in this scenario, we have $\hat{\mathcal{L}}(\boldsymbol{\psi}) = \mathcal{L}(\boldsymbol{\theta})$ whenever $\boldsymbol{\psi} = \varphi(\boldsymbol{\theta})$, it is well-known that downstream quantities such as optimization trajectories [55, 71], the Hessian [20, Sec. 5], and probability densities over $\boldsymbol{\theta}$ [59, Sec. 5.2.1.4] are not invariant under model reparametrization. These non-invariances are detrimental since an arbitrary reparametrization can affect the studied quantities and thus consistency cannot be guaranteed—parametrization muddles the analysis.

For instance, because of this, (i) one cannot relate Hessian-based sharpness measures with generalization and the correlation analyses between them [27, 33, 73] become meaningless, (ii) a good preconditioner cannot be guaranteed to be good anymore for optimizing the reparametrized model—this is one reason why invariant methods like natural gradient are desirable [37, 50, 58, 62], and (iii) under a reparametrization, a prior density might put low probability in a region that corresponds to the high-probability region in the original space, making posterior inference pathological [25].

To analyze this issue, in this work we adopt the framework of Riemannian geometry, which is a generalization of calculus studying the intrinsic properties of manifolds. "Intrinsic" in this context means that objects such as functions, vectors, and tensors defined on the manifold must be independent of how the manifold is represented via a coordinate system, and conserved under a change of coordinates [47]. The parameter space of a neural network, which is by default assumed to be $\Theta = \mathbb{R}^d$ with the Euclidean metric and the Cartesian coordinates, is a Riemannian manifold—model reparametrization is such a change in the coordinate system.

Why, then, can reparametrizations bring up the aforementioned inconsistencies? In this work, we discuss this discrepancy. We observe that this issue often arises because the Riemannian metric is left implicit, and dropped when computing downstream quantities such as gradients, Hessians, and volumes on the parameter space. This directly suggests the solution: Invariance under reparametrization is guaranteed if the Riemannian metric is not just implicitly assumed, but made explicit, and it is made sure that the associated transformation rules of objects such as vectors, covectors, and tensors are performed throughout. We show how these insights apply to common cases (Fig. 2).

**Limitation** Our work focuses on the *reparametrization consistency* of prior and future methods that leverage geometric objects such as gradients and the Hessian. Thus, we leave the geometric analysis for invariance under symmetry as future work. In any case, our work is complimentary to other works that analyze invariance under symmetry [17, 22, 43, 67, etc]. Moreover, this work's focus is not to propose "better" methods e.g. for better preconditioners or better generalization metrics. Rather, we provide a guardrail for existing and future methods to avoid pathologies due to reparametrization.

## 2 Preliminaries

This section provides background and notation on relevant concepts of neural networks and Riemannian geometry. For the latter, we frame the discussion in terms of linear algebra as close as possible to the notation of the ML community: We use regular faces to denote abstract objects and bold faces for

their concrete representations in a particular parametrization. E.g., a linear map $A$ is represented by a matrix $\boldsymbol{A}$, a point $z$ is represented by a tuple of numbers $\boldsymbol{\theta}$, an inner product $G(z)$ at $z$ is represented by a matrix $\boldsymbol{G}(\boldsymbol{\theta})$. The same goes for the differential $\nabla$, $\boldsymbol{\nabla}$; and the gradient $\mathrm{grad}$, $\mathbf{grad}$.

## 2.1  Some notation for neural networks

The following concepts are standard, introduced mostly to clarify notation. Let $f : \mathbb{R}^n \times \mathbb{R}^d \to \mathbb{R}^k$ with $(\boldsymbol{x}, \boldsymbol{\theta}) \mapsto f(\boldsymbol{x}; \boldsymbol{\theta})$ be a model with input, output, and parameter spaces $\mathbb{R}^n$, $\mathbb{R}^k$, $\mathbb{R}^d$. Let $\mathcal{D} := \{(\boldsymbol{x}_i, \boldsymbol{y}_i)\}_{i=1}^m$ be a dataset and write $\boldsymbol{X} := \{\boldsymbol{x}_i\}_{i=1}^m$ and $\boldsymbol{Y} := \{\boldsymbol{y}_i\}_{i=1}^m$. We moreover write $f(\boldsymbol{X}; \boldsymbol{\theta}) := \mathrm{vec}(\{f(\boldsymbol{x}_i; \boldsymbol{\theta})\}_{i=1}^m) \in \mathbb{R}^{mk}$. The standard way of training $f$ is by finding a point $\boldsymbol{\theta}^* \in \mathbb{R}^d$ s.t. $\boldsymbol{\theta}^* = \arg\min_{\boldsymbol{\theta} \in \mathbb{R}^d} \sum_{i=1}^m \ell(\boldsymbol{y}_i, f(\boldsymbol{x}_i; \boldsymbol{\theta})) =: \arg\min_{\boldsymbol{\theta} \in \mathbb{R}^d} \mathcal{L}(\boldsymbol{\theta})$, for some loss function $\ell$. If we add a weight decay $\gamma/2\|\boldsymbol{\theta}\|_2^2$ term to $\mathcal{L}(\boldsymbol{\theta})$, the minimization problem has a probabilistic interpretation as *maximum a posteriori* (MAP) estimation of the posterior density $p(\boldsymbol{\theta} \mid \mathcal{D})$ under the likelihood function $p(\boldsymbol{Y} \mid f(\boldsymbol{X}; \boldsymbol{\theta})) \propto \exp(-\sum_{i=1}^m \ell(\boldsymbol{y}_i, f(\boldsymbol{x}_i; \boldsymbol{\theta})))$ and an isotropic Gaussian prior $p(\boldsymbol{\theta}) = \mathcal{N}(\boldsymbol{\theta} \mid \boldsymbol{0}, \gamma^{-1}\boldsymbol{I})$. We denote the MAP loss by $\mathcal{L}_{\mathrm{MAP}}$. In our present context, this interpretation is relevant because this probabilistic interpretation implies a probability density, and it is widely known that a probability density transforms nontrivially under reparametrization.

A textbook way of obtaining $\boldsymbol{\theta}^*$ is gradient descent (GD): At each time step $t$, obtain the next estimate $\boldsymbol{\theta}^{(t+1)} = \boldsymbol{\theta}^{(t)} - \alpha \boldsymbol{\nabla}\mathcal{L}|_{\boldsymbol{\theta}^{(t)}}$, for some $\alpha > 0$. This can be considered as the discretization of the gradient flow ODE $\dot{\boldsymbol{\theta}} = -\boldsymbol{\nabla}\mathcal{L}|_{\boldsymbol{\theta}}$. Among the many variants of GD is preconditioned GD which considers a positive-definite matrix field $\boldsymbol{R}$ on $\mathbb{R}^d$, yielding $\dot{\boldsymbol{\theta}} = -\boldsymbol{R}(\boldsymbol{\theta})^{-1}\boldsymbol{\nabla}\mathcal{L}|_{\boldsymbol{\theta}}$.

## 2.2  The geometry of the parameter space

**Remark 1.** We restrict our discussion to global coordinate charts since they are the default assumption in neural networks, and in a bid to make our discussion clear for people outside of the differential geometry community. We refer the curious reader to Appendix D for a discussion on local coordinate charts. Note that, our results hold in the general local-chart formulation. ∎

The parameter space $\mathbb{R}^d$ is the canonical example of a smooth manifold. We can impose upon this manifold a (global) coordinate chart: a homeomorphism that represents a point with an element of $\mathbb{R}^d$. The standard choice is the Cartesian coordinate system $\theta : \mathbb{R}^d \to \Theta \cong \mathbb{R}^d$, which we have used in the previous section. That is, the image of $\theta$ uniquely represents elements of $\mathbb{R}^d$—given a point $z \in \mathbb{R}^d$, we write $\boldsymbol{\theta} = \theta(z)$ for its coordinate representation under $\theta$.

The choice of a coordinate system is not unique. Any homeomorphism $\psi : \mathbb{R}^d \to \Psi \cong \mathbb{R}^d$ can be used as an alternative coordinate chart. For instance, the polar coordinates can be used to represent points in $\mathbb{R}^d$ instead. Crucially, the images of any pair of coordinates must be con-

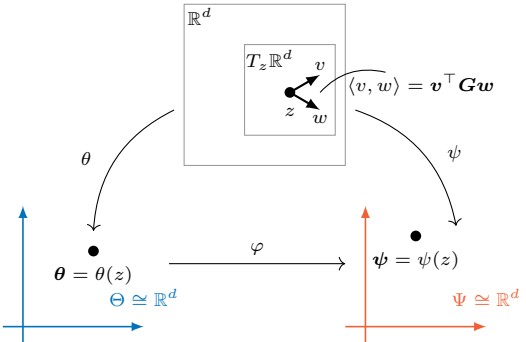

Figure 3: The implicit geometric assumption on the parameter space $\mathbb{R}^d$. $\theta, \psi$ are two different (global) coordinates on $\mathbb{R}^d$.

nected through a diffeomorphism (a smooth function with a smooth inverse) $\varphi : \Theta \to \Psi$. Such a map is called a ***change of coordinates*** or ***reparametrization*** since it acts on the parameter space.

**Example 2 (Reparametrization).** Reparametrization is ubiquitous in machine learning:

    (a) *Mean-Field Variational Inference.* Let $q(\boldsymbol{x}; \boldsymbol{\theta})$ be a variational approximation, which is often chosen to be $\mathcal{N}(\boldsymbol{x} \mid \boldsymbol{\mu}, \mathrm{diag}(\boldsymbol{\sigma}^2))$, i.e. $\boldsymbol{\theta} = \{\boldsymbol{\mu} \in \mathbb{R}^d, \boldsymbol{\sigma}^2 \in \mathbb{R}_{>0}^d\}$. Common choices of reparametrization of $\boldsymbol{\sigma}^2$ include the log-space [52] or softplus [12] parametrization.

    (b) *Weight normalization.* Given a NN $f_{\boldsymbol{\theta}}$, WeightNorm [69] applies the reparametrization $\boldsymbol{\psi} = r\boldsymbol{\eta}$ where $r \in \mathbb{R}_{>0}$ and $\boldsymbol{\eta} := \boldsymbol{\theta}/\|\boldsymbol{\theta}\| \in \mathbb{S}^{d-1}$. This is akin to the polar coordinates. (Note that we assume $\boldsymbol{\theta} \in \mathbb{R}^d \setminus \{\boldsymbol{0}\}$ since otherwise $\varphi$ is not a global diffeomorphism.) ∎

At each point $z \in \mathbb{R}^d$, there exists a vector space $T_z\mathbb{R}^d$ called the ***tangent space at*** $z$, consisting of the so-called ***tangent vectors***. An important example of a tangent vector is the gradient vector $\mathrm{grad}\,\mathcal{L}|_z$

of $\mathcal{L} : \mathbb{R}^d \to \mathbb{R}$ at $z$. The dual space $T_z^*\mathbb{R}^d$ of $T_z\mathbb{R}^d$ is referred to as the ***cotangent space*** and consists of linear functionals of $T_z\mathbb{R}^d$, called ***tangent covectors***. An example of a tangent covector is the differential $\nabla\mathcal{L}|_z$ of $\mathcal{L}$ at $z$. Under a coordinate system, one can think of both tangent vectors and covectors as vectors in the sense of linear algebra, i.e., tuples of numbers.

One can take an inner product of tangent vectors by equipping the manifold with a ***Riemannian metric*** $G$, which, at each point in $\mathbb{R}^d$ is represented by a positive-definite $d \times d$ matrix $G$ whose coefficients depend on the choice of coordinates. With the help of the metric, there is an isomorphism between $T_z\mathbb{R}^d$ and $T_z^*\mathbb{R}^d$. In coordinates, it is given by the map $T_z\mathbb{R}^d \to T_z^*\mathbb{R}^d : \boldsymbol{v} \mapsto \boldsymbol{G}\boldsymbol{v}$ and its inverse $T_z^*\mathbb{R}^d \to T_z\mathbb{R}^d : \boldsymbol{\omega} \mapsto \boldsymbol{G}^{-1}\boldsymbol{\omega}$. One important instance of this isomorphism is the fact that $\operatorname{grad}\mathcal{L}|_z$ is represented by $\boldsymbol{G}^{-1}\boldsymbol{\nabla}\mathcal{L}$ in coordinates. The natural gradient is a direct consequence of this fact. In practice, it is standard to assume the Cartesian coordinates on $\mathbb{R}^d$, implying $\boldsymbol{G} \equiv \boldsymbol{I}$. We have thus $\operatorname{\mathbf{grad}}\mathcal{L} \equiv \boldsymbol{\nabla}\mathcal{L}$, which only seems trivial at first sight, but reveals the relationship between the tangent vector on the l.h.s. and the cotangent vector on the r.h.s.

**Remark 3.** Another common way to define a NN's parameter space is to define a manifold of probabilistic models $M := \{p(\boldsymbol{Y} \mid f(\boldsymbol{X};\boldsymbol{\theta})) : \boldsymbol{\theta} \in \mathbb{R}^d\}$ and assume that $p(\boldsymbol{Y} \mid f(\boldsymbol{X};\boldsymbol{\theta})) \mapsto \boldsymbol{\theta}$ is the coordinate chart [19, 55, etc.]. The problem with this construction is that there is no bijection $\mathbb{R}^d \to M$ in general. Indeed, the Jacobian of the map $f(\boldsymbol{X}; \cdot) : \mathbb{R}^d \to M$ is in practice surjective everywhere due to overparametrization [34, 78]. Therefore, one cannot define a proper metric on the parameter space $\mathbb{R}^d$ that corresponds to a metric on the distribution space $M$ [46, Prop. 13.9]. For instance, using this construction, the Fisher metric is singular for overparametrized NNs [10, 34, 74, 78] and thus not a valid Riemannian metric. The pseudoinverse has been used to handle this but is mostly limited to theoretical analyses [10, 74]: As far as we know, in practice, damping—which breaks the interpretation of the Fisher on $\mathbb{R}^d$ as the pullback metric from $M$—is *de facto* for handling this due to its numerical stability [4].

By detaching from the distribution space, the definition used in this work does not have this issue. It enables more general constructions of metrics in the parameter space since *any* positive-definite matrix is admissible. E.g. one is free to add damping or use any approximation to the Fisher—our results still apply to this case. Thus, our construction is closer to practice. ∎

### 2.2.1 Transformation rules

Differential geometry is the study of coordinate-independent objects: Geometric objects must be *invariant* under change-of-coordinates in the sense that any pair of representations must refer to the same (abstract) object. Suppose $\varphi : \Theta \to \Psi$ is a reparametrization, with $\boldsymbol{\psi} = \varphi(\boldsymbol{\theta})$. Coordinate independence is encoded in the following transformation rules, which involve the Jacobian $\boldsymbol{J}(\boldsymbol{\theta}) = (\partial\boldsymbol{\psi}_i/\partial\boldsymbol{\theta}_j)$ of $\varphi$, and its inverse $\boldsymbol{J}^{-1}(\boldsymbol{\psi}) = (\boldsymbol{J}(\boldsymbol{\theta}))^{-1} = (\partial\boldsymbol{\theta}_i/\partial\boldsymbol{\psi}_j)$—the Jacobian of $\varphi^{-1}$. (Color codes are used for clarity when different objects are present in a single expression later on.)

(a) A function $h : \Theta \to \mathbb{R}$ in $\theta$-coordinates transforms into $\hat{h} = h \circ \varphi^{-1}$ in $\psi$-coordinates.

(b) A tangent vector $\boldsymbol{v}$ in $\theta$-coordinates transforms into $\boldsymbol{J}(\boldsymbol{\theta})\boldsymbol{v}$. In particular, a gradient vector $\operatorname{\mathbf{grad}}\mathcal{L}|_{\boldsymbol{\theta}}$ transforms into $\boldsymbol{J}(\boldsymbol{\theta})\operatorname{\mathbf{grad}}\mathcal{L}|_{\boldsymbol{\theta}}$.

(c) A tangent covector $\boldsymbol{\omega}$ in $\theta$-coordinates transforms into $\boldsymbol{J}^{-1}(\boldsymbol{\psi})^\top\boldsymbol{\omega}$. In particular, given a transformed function $\mathcal{L} \circ \varphi^{-1} : \Psi \to \mathbb{R}$, we have $\boldsymbol{\nabla}(\mathcal{L} \circ \varphi^{-1})|_{\boldsymbol{\psi}} = \boldsymbol{J}^{-1}(\boldsymbol{\psi})^\top\boldsymbol{\nabla}\mathcal{L}|_{\varphi^{-1}(\boldsymbol{\psi})}$, which we recognize as the standard chain rule.

(d) A metric $\boldsymbol{G}(\boldsymbol{\theta})$ becomes $\boldsymbol{J}(\boldsymbol{\theta})^{-\top}\boldsymbol{G}(\boldsymbol{\theta})\boldsymbol{J}(\boldsymbol{\theta})^{-1} = \boldsymbol{J}^{-1}(\boldsymbol{\psi})^\top\boldsymbol{G}(\boldsymbol{\psi})\boldsymbol{J}^{-1}(\boldsymbol{\psi})$. In general, this rule applies to any tensor that takes tangent vectors as its arguments, e.g. a bilinear map.

The following examples show how these rules ensure the invariance of geometric objects.

**Example 4.** Let $h, \boldsymbol{v}, \boldsymbol{\omega}, \boldsymbol{G}$ respectively be a function, vector, covector, and metric in $\theta$-coordinates at a point $\boldsymbol{\theta} = \theta(z)$, and let $\varphi : \Theta \to \Psi$ be a reparametrization.

(a) Let $\boldsymbol{\psi} := \varphi(\boldsymbol{\theta})$ and $\hat{h} := h \circ \varphi^{-1}$ be $\boldsymbol{\theta} \in \Theta$ and $h$ expressed in $\psi$-coordinates, respectively. Then, $\hat{h}(\boldsymbol{\psi}) = h(\varphi^{-1}(\boldsymbol{\psi})) = h(\boldsymbol{\theta})$. That is, the actions of $h$ and $\hat{h}$ agree in both coordinates and thus they represent the same abstract function on $\mathbb{R}^d$.

(b) The action of $\boldsymbol{\omega}$ on $\boldsymbol{v}$ in $\theta$-coordinates is given by the product $\boldsymbol{\omega}^\top\boldsymbol{v}$. Let $\hat{\boldsymbol{v}} := \boldsymbol{J}(\boldsymbol{\theta})\boldsymbol{v}$ and $\hat{\boldsymbol{\omega}} := \boldsymbol{J}(\boldsymbol{\theta})^{-\top}\boldsymbol{\omega}$ be the transformed vector and covectors. Then,

$$\hat{\boldsymbol{\omega}}^\top\hat{\boldsymbol{v}} = (\boldsymbol{J}(\boldsymbol{\theta})^{-\top}\boldsymbol{\omega})^\top(\boldsymbol{J}(\boldsymbol{\theta})\boldsymbol{v}) = \boldsymbol{\omega}^\top\boldsymbol{v}. \tag{1}$$

That is, both $\omega$ and $\hat{\omega}$ are the representations of the same linear functional; $v$ and $\hat{v}$ are the representations of the same tangent vector.

(c) Let $\hat{G} := J^{-\top} G J^{-1}$ be the transformed metric in the $\psi$-coordinates. Then,

$$\hat{v}^\top \hat{G} \hat{v} = (Jv)^\top J^{-\top} G J^{-1} (Jv) = v^\top G v,$$

and thus the transformation rules ensure that inner products are also invariant. $\blacksquare$

Finally, we say that an ODE's dynamics is invariant if the trajectory in parametrization corresponds to the trajectory in another parametrization. Concretely, a trajectory $(\boldsymbol{\theta}_t)_t$ in $\Theta$ is invariant if under the reparametrization $\varphi : \Theta \to \Psi$, the transformed trajectory $(\boldsymbol{\psi}_t)_t$ is related to $(\boldsymbol{\theta}_t)_t$ by $\boldsymbol{\theta}_t = \varphi^{-1}(\boldsymbol{\psi}_t)$ for each $t$. See Fig. 2a for an illustration.

## 3  Neural Networks and Reparametrization

We discuss three aspects of the parameter space under reparametrization, as illustrated in Fig. 2. First, we address the non-invariance of Hessian-based flatness measures [e.g., 20], and show how taking the metric into account provides invariance. Second, we show that the invariance property often cited in favor of natural gradient is not unique, but an inherent property of any gradient descent algorithm when considering its ODE. Finally, we show that modes of probability density functions on the parameter space are invariant when the Lebesgue measure is generalized using the metric.

### 3.1  Invariance of the Hessian

A growing body of literature connects the flatness of minima found by optimizers to generalization performance [8, 13, 23, 27, 36, 38, 48, 53]. However, as Dinh et al. [20] observed, this association does not have a solid foundation since standard sharpness measures derived from the Hessian of the loss function are not invariant under reparametrization.

From the Riemannian-geometric perspective, the Hessian of a function $\mathcal{L}$ (or $\mathcal{L}_{\mathrm{MAP}}$ or any other twice-differentiable function) on $\theta$-coordinates is represented by a $d \times d$ matrix with coefficients $\boldsymbol{H}_{ij}(\boldsymbol{\theta}) := \frac{\partial^2 \mathcal{L}}{\partial \boldsymbol{\theta}_i \partial \boldsymbol{\theta}_j}(\boldsymbol{\theta}) - \sum_{k=1}^{d} \boldsymbol{\Gamma}_{ij}^k(\boldsymbol{\theta}) \frac{\partial \mathcal{L}}{\partial \boldsymbol{\theta}_k}(\boldsymbol{\theta})$, for any $\boldsymbol{\theta} \in \Theta$ and $i, j = 1, \ldots, d$, where $\boldsymbol{\Gamma}_{ij}^k$ is a particular three-dimensional array. While it might seem daunting, when $\boldsymbol{\theta}$ is a local minimum of $\mathcal{L}$, the second term is zero since the partial derivatives of $\mathcal{L}$ are all zero at $\boldsymbol{\theta}$. Thus, $\boldsymbol{H}(\boldsymbol{\theta})$ equals the standard Hessian matrix $(\partial^2 \mathcal{L}/\partial \boldsymbol{\theta}_i \partial \boldsymbol{\theta}_j)$ at $\boldsymbol{\theta} \in \arg\min \mathcal{L}$.

Considering the Hessian as a bilinear function that takes two vectors and produces a number, it follows the covariant transformation rule, just like the metric (see Appendix B.1 for a derivation): Let $\varphi : \Theta \to \Psi$ and $\boldsymbol{\psi} = \varphi(\boldsymbol{\theta})$. The Hessian matrix $\boldsymbol{H}(\boldsymbol{\theta})$ at a local minimum in $\theta$-coordinates transforms into $\hat{\boldsymbol{H}}(\boldsymbol{\psi}) = \boldsymbol{J}^{-1}(\boldsymbol{\psi})^\top \boldsymbol{H}(\varphi^{-1}(\boldsymbol{\psi})) \boldsymbol{J}^{-1}(\boldsymbol{\psi})$ in $\psi$-coordinates—this is correctly computed by automatic differentiation, i.e. by chain and product rules. But, while this gives invariance of $\boldsymbol{H}$ as a bilinear map (Example 4c), the determinant of $\boldsymbol{H}$—a popular flatness measure—is not invariant because

$$(\det \hat{\boldsymbol{H}})(\boldsymbol{\psi}) := \det \left( \boldsymbol{J}^{-1}(\boldsymbol{\psi})^\top \boldsymbol{H}(\varphi^{-1}(\boldsymbol{\psi})) \boldsymbol{J}^{-1}(\boldsymbol{\psi}) \right) = (\det \boldsymbol{J}^{-1}(\boldsymbol{\psi}))^2 (\det \boldsymbol{H}(\varphi^{-1}(\boldsymbol{\psi}))),$$

and so in general, we do not have the relation $(\det \hat{\boldsymbol{H}}) = (\det \boldsymbol{H}) \circ \varphi^{-1}$ that would make this function invariant under reparametrization (Example 4a).

The key to obtaining invariance is to employ the metric $\boldsymbol{G}$ to transform the bilinear Hessian into a linear map/operator on the tangent space. This is done by simply multiplying $\boldsymbol{H}$ with the inverse of the metric $\boldsymbol{G}$, i.e. $\boldsymbol{E} := \boldsymbol{G}^{-1} \boldsymbol{H}$. The determinant of $\boldsymbol{E}$ is thus an invariant quantity.[1] To show this, under $\varphi$, the linear map $\boldsymbol{E}$ transforms into

$$\begin{aligned}
\hat{\boldsymbol{E}}(\boldsymbol{\psi}) &= (\boldsymbol{J}^{-1}(\boldsymbol{\psi})^\top \boldsymbol{G}(\varphi^{-1}(\boldsymbol{\psi})) \boldsymbol{J}^{-1}(\boldsymbol{\psi}))^{-1} \boldsymbol{J}^{-1}(\boldsymbol{\psi})^\top \boldsymbol{H}(\varphi^{-1}(\boldsymbol{\psi})) \boldsymbol{J}^{-1}(\boldsymbol{\psi}) \\
&= (\boldsymbol{J}^{-1}(\boldsymbol{\psi}))^{-1} \boldsymbol{G}(\varphi^{-1}(\boldsymbol{\psi})) \boldsymbol{H}(\varphi^{-1}(\boldsymbol{\psi})) \boldsymbol{J}^{-1}(\boldsymbol{\psi}),
\end{aligned} \tag{2}$$

due to the transformation of both $\boldsymbol{G}$ and $\boldsymbol{H}$. Hence, $\det \boldsymbol{E}$ transforms into

$$(\det \hat{\boldsymbol{E}})(\boldsymbol{\psi}) = (\det \boldsymbol{J}^{-1}(\boldsymbol{\psi}))^{-1} (\det \boldsymbol{J}^{-1}(\boldsymbol{\psi})) \det(\boldsymbol{G}(\varphi^{-1}(\boldsymbol{\psi})) \boldsymbol{H}(\varphi^{-1}(\boldsymbol{\psi}))) = (\det \boldsymbol{E})(\varphi^{-1}(\boldsymbol{\psi}))$$

---

[1]This is because one can view the loss landscape as a hypersurface: $\boldsymbol{H}$ is connected to the *second fundamental form* and $\boldsymbol{E}$ to the *shape operator*. $\det \boldsymbol{E}$ is thus related to the invariant *Gaussian curvature* [47, Ch. 8].

in $\psi$-coordinates. Thus, we have the desired invariant transformation $(\det \hat{\boldsymbol{E}}) = (\det \boldsymbol{E}) \circ \varphi^{-1}$. Note that $\boldsymbol{G}$ is an arbitrary metric—this invariance thus also holds for the Euclidean case where $\boldsymbol{G} \equiv \boldsymbol{I}$. Note further that the trace and eigenvalues of $\boldsymbol{E}$ are also invariant; see Appendices B.2 and B.3. Finally, see Appendix A for a simple working example.

---

To obtain invariance in the Hessian-determinant, -trace, and -eigenvalues at a minimum of $\mathcal{L}$, we explicitly write it as $\boldsymbol{E} = \boldsymbol{G}^{-1}\boldsymbol{H}$, even when $\boldsymbol{G} \equiv \boldsymbol{I}$, and transform it according to Section 2.2.1.

---

## 3.2 Invariance of gradient descent

Viewed from the geometric perspective, both gradient descent (GD) and natural gradient descent (NGD) come from the same ODE framework $\dot{\boldsymbol{\theta}} = -\boldsymbol{G}(\boldsymbol{\theta})^{-1}\boldsymbol{\nabla}\mathcal{L}|_\theta$. But NGD is widely presented as invariant under reparametrization, while GD is not [51, 71, etc.]. Is the choice of the metric $\boldsymbol{G}$ the cause? Here we will show from the framework laid out in Section 2.2 that *any* metric is invariant *if* its transformation rule is faithfully followed. And thus, the invariance of NGD is not unique. Rather, the Fisher metric used in NGD is part of the family of metrics that transform correctly under autodiff, and thus it is "automatically" invariant under standard deep learning libraries like PyTorch, TensorFlow, and JAX [1, 24, 66]—see Appendix C.

The underlying assumption of GD is that one works in the Cartesian coordinates and that $\boldsymbol{G} \equiv \boldsymbol{I}$. For this reason, one can ignore the metric $\boldsymbol{G}$ in $\dot{\boldsymbol{\theta}} = -\boldsymbol{G}(\boldsymbol{\theta})^{-1}\boldsymbol{\nabla}\mathcal{L}|_\theta$, and simply write $\dot{\boldsymbol{\theta}} = -\boldsymbol{\nabla}\mathcal{L}|_\theta$. This is correct but this simplification is exactly what makes GD not invariant under a reparametrization $\varphi : \Theta \to \Psi$ where $\boldsymbol{\psi} = \varphi(\boldsymbol{\theta})$. To see this, notice that while GD transforms $\boldsymbol{\nabla}\mathcal{L}$ correctly via the chain rule $\dot{\boldsymbol{\psi}} = -\boldsymbol{J}^{-1}(\boldsymbol{\psi})^\top \boldsymbol{\nabla}\mathcal{L}|_{\varphi^{-1}(\boldsymbol{\psi})}$, by ignoring the metric $\boldsymbol{I}$, one would miss the important fact that it must also be transformed into $\hat{\boldsymbol{G}}(\boldsymbol{\psi}) = \boldsymbol{J}^{-1}(\boldsymbol{\psi})^\top \boldsymbol{J}^{-1}(\boldsymbol{\psi})$. It is clear that $\hat{\boldsymbol{G}}(\boldsymbol{\psi})$ does not equal $\boldsymbol{I}$ in general. Thus, we cannot ignore this term in the transformed dynamics since it would imply that one uses a different metric—the dynamics are thus different in the $\theta$- and $\psi$-coordinates. When one explicitly considers the above transformation, one obtains

$$\dot{\boldsymbol{\psi}} = -\hat{\boldsymbol{G}}(\boldsymbol{\psi})^{-1}\boldsymbol{J}^{-1}(\boldsymbol{\psi})^\top \boldsymbol{\nabla}\mathcal{L}|_{\varphi^{-1}(\boldsymbol{\psi})} = -(\boldsymbol{J}^{-1}(\boldsymbol{\psi})^\top \boldsymbol{J}^{-1}(\boldsymbol{\psi}))^{-1}\boldsymbol{J}^{-1}(\boldsymbol{\psi})^\top \boldsymbol{\nabla}\mathcal{L}|_{\varphi^{-1}(\boldsymbol{\psi})}$$
$$= -\boldsymbol{J}(\varphi^{-1}(\boldsymbol{\psi}))\boldsymbol{\nabla}\mathcal{L}|_{\varphi^{-1}(\boldsymbol{\psi})}.$$

This dynamics in $\psi$-coordinates preserves the assumption that the metric is $\boldsymbol{I}$ in $\theta$-coordinates and thus invariant. In contrast, the dynamics $\dot{\boldsymbol{\psi}}$ under just the chain rule implicitly changes the metric in $\theta$-coordinates, i.e. from $\boldsymbol{I}$ into $\boldsymbol{J}(\boldsymbol{\theta})^\top \boldsymbol{J}(\boldsymbol{\theta})$, and thus the trajectories are not invariant.

This discussion can be extended to *any* metric $\boldsymbol{R}$: Simply use the transformed metric $\hat{\boldsymbol{R}}(\boldsymbol{\psi}) = \boldsymbol{J}^{-1}(\boldsymbol{\psi})^\top \boldsymbol{R}(\varphi^{-1}(\boldsymbol{\psi}))\boldsymbol{J}^{-1}(\boldsymbol{\psi})$, and we obtain the invariant dynamics of any preconditioned GD with preconditioner $\boldsymbol{R}$ given by $\dot{\boldsymbol{\psi}} = -\boldsymbol{J}(\varphi^{-1}(\boldsymbol{\psi}))\boldsymbol{R}(\varphi^{-1}(\boldsymbol{\psi}))^{-1}\boldsymbol{\nabla}\mathcal{L}|_{\varphi^{-1}(\boldsymbol{\psi})}$.

---

To obtain invariance in optimizers with any metric/preconditioner, explicitly write down the metric even if it is trivial, and perform the proper transformation under reparametrization.

---

**Remark 5.** For the discretized dynamics, the larger the step size, the less exact the invariance. For instance, *discrete* natural gradient update rule is only invariant up to the first-order [50, 71]. This is orthogonal to our work since it is about improving ODE solvers [71]. ∎

The consequence is that we need a "geometric-aware" autodiff library such that the invariant dynamics above is always satisfied. In this case, *any* preconditioner $\boldsymbol{R}$ yields invariance under *any* reparametrization, even nonlinear ones. This is in contrast to the current literature, e.g. under standard autodiff, Newton's method is only affine-invariant, and structured approximate NGD methods such as K-FAC are only invariant under a smaller class of reparametrizations [56].

## 3.3 Invariance in probability densities

Let $q_\Theta(\boldsymbol{\theta})$ be a probability density function (pdf) under the Lebesgue measure $d\boldsymbol{\theta}$ on $\Theta$. Under a reparametrization $\varphi : \Theta \to \Psi$ with $\boldsymbol{\psi} = \varphi(\boldsymbol{\theta})$, it transforms into $q_\Psi(\boldsymbol{\psi}) = q_\Theta(\varphi^{-1}(\boldsymbol{\psi}))|\det \boldsymbol{J}^{-1}(\boldsymbol{\psi})|$. This transformation rule ensures $q_\Psi$ to be a valid pdf under the Lebesgue measure $d\boldsymbol{\psi}$ on $\Psi$, i.e. $\int_\Psi q_\Psi(\boldsymbol{\psi})\,d\boldsymbol{\psi} = 1$. Notice, in general $q_\Psi \neq q_\Theta \circ \varphi^{-1}$ due to the change-of-random-variable rule. Hence, pdfs transform differently than standard functions (Example 4a)

and can thus have non-invariant modes [cf. 59, Sec. 5.2.1.4]: Density maximization, such as the optimization of $\mathcal{L}_{\text{MAP}}$, is not invariant even if an invariant optimizer is employed.

Just like the discussion in the previous section, it is frequently suggested that to obtain invariance here, one must again employ the help of the Fisher matrix $\boldsymbol{F}$ [21]. When applied to a prior, this gives rise to the famous *Jeffreys [31] prior* $p_J(\boldsymbol{\theta}) \propto |\det \boldsymbol{F}(\boldsymbol{\theta})|^{\frac{1}{2}}$ with normalization constant $\int_\Theta |\det \boldsymbol{F}(\boldsymbol{\theta})|^{\frac{1}{2}} d\boldsymbol{\theta}$. Is the Fisher actually necessary to obtain invariance in pdf maximization?

Here, we show that the same principle of "being aware of the implicit metric and following proper transformation rules" can be applied. A pdf $q_\Theta(\boldsymbol{\theta})$ can be written as $q_\Theta(\boldsymbol{\theta}) = \frac{q_\Theta(\boldsymbol{\theta})\, d\boldsymbol{\theta}}{d\boldsymbol{\theta}}$ to explicitly show the dependency of the base measure $d\boldsymbol{\theta}$—this is the Lebesgue measure on $\Theta$, which is the natural unit-volume[2] measure in Euclidean space. Given a metric $\boldsymbol{G}$, the natural volume-measurement device on $\Theta$ is the ***Riemannian volume form*** $dV_G$ which has $\theta$-coordinate representation $dV_{\boldsymbol{G}} := |\det \boldsymbol{G}(\boldsymbol{\theta})|^{\frac{1}{2}} d\boldsymbol{\theta}$. This object takes the role of the Lebesgue measure on a Riemannian manifold: Intuitively, it behaves like the Lebesgue measure but takes the (local) distortion due to the metric $\boldsymbol{G}$ into account. Indeed, when $\boldsymbol{G} \equiv \boldsymbol{I}$ we recover $d\boldsymbol{\theta}$.

Here, we instead argue that invariance is an inherent property of the modes of probability densities, as long as we remain aware of the metric and transform it properly under reparametrization. Explicitly acknowledging the presence of the metric, we obtain

$$\frac{q_\Theta(\boldsymbol{\theta})\, d\boldsymbol{\theta}}{|\det \boldsymbol{G}(\boldsymbol{\theta})|^{\frac{1}{2}} d\boldsymbol{\theta}} = q_\Theta(\boldsymbol{\theta})\, |\det \boldsymbol{G}(\boldsymbol{\theta})|^{-\frac{1}{2}} =: q_\Theta^{\boldsymbol{G}}(\boldsymbol{\theta}). \tag{3}$$

This is a valid probability density under $dV_G$ on $\Theta$ since $\int_\Theta q_\Theta^{\boldsymbol{G}}(\boldsymbol{\theta})\, dV_{\boldsymbol{G}} = 1$. This formulation generalizes the standard Lebesgue density. And, it becomes clear that the Jeffreys prior is simply the uniform distribution under $dV_{\boldsymbol{F}}$; its density is $q_\Theta^{\boldsymbol{F}} \equiv 1$ under $dV_{\boldsymbol{F}}$.

We can now address the invariance question. Under $\varphi$, considering the transformation rules for both $q_\Theta(\boldsymbol{\theta})$ and $\boldsymbol{G}$, the density (3) thus becomes

$$
\begin{aligned}
q_\Psi^{\boldsymbol{G}}(\boldsymbol{\psi}) &= q_\Theta(\varphi^{-1}(\boldsymbol{\psi}))\, |\det \boldsymbol{J}^{-1}(\boldsymbol{\psi})|\, |\det(\boldsymbol{J}^{-1}(\boldsymbol{\psi})^\top \boldsymbol{G}(\varphi^{-1}(\boldsymbol{\psi}))\boldsymbol{J}^{-1}(\boldsymbol{\psi}))|^{-\frac{1}{2}} \\
&= q_\Theta(\varphi^{-1}(\boldsymbol{\psi}))\, |\det \boldsymbol{G}(\varphi^{-1}(\boldsymbol{\psi}))|^{-\frac{1}{2}} = q_\Theta^{\boldsymbol{G}}(\varphi^{-1}(\boldsymbol{\psi})).
\end{aligned}
\tag{4}
$$

This means, $q_\Theta^{\boldsymbol{G}}$ transforms into $q_\Theta^{\boldsymbol{G}} \circ \varphi^{-1}$ and is thus invariant since it transforms as standard function—notice the lack of the Jacobian-determinant term here, compared to the standard change-of-density rule. In particular, just like standard unconstrained functions, the modes of $q_\Theta^{\boldsymbol{G}}$ are invariant under reparametrization. Since $\boldsymbol{G}$ is arbitrary, this also holds for $\boldsymbol{G} \equiv \boldsymbol{I}$, and thus the modes of Lebesgue-densities are invariant, as long as $\boldsymbol{I}$ is transformed correctly (Fig. 2b).

Note that, even if the transformation rule is now different from the one in standard probability theory, $q_\Psi^{\boldsymbol{G}}$ is a valid density under the transformed volume form. This is because due to the transformation of the metric $\boldsymbol{G} \mapsto \hat{\boldsymbol{G}}$, we have $dV_{\hat{\boldsymbol{G}}} = |\det \boldsymbol{G}(\varphi^{-1}(\boldsymbol{\psi}))|^{\frac{1}{2}} |\det \boldsymbol{J}^{-1}(\boldsymbol{\psi})|\, d\boldsymbol{\psi}$. Thus, together with (4), we have $\int_\Psi q_\Psi^{\boldsymbol{G}}(\boldsymbol{\psi})\, dV_{\hat{\boldsymbol{G}}} = 1$. This also shows that the $|\det \boldsymbol{J}^{-1}(\boldsymbol{\psi})|$ term in the standard change-of-density formula (i.e. when $\boldsymbol{G} \equiv \boldsymbol{I}$) is actually part of the transformation of $d\boldsymbol{\theta}$.

Put another way, the standard change-of-density formula ignores the metric. The resulting density thus must integrate w.r.t. $d\boldsymbol{\psi}$—essentially assuming a change of geometry, not just a simple reparametrization—and thus $q_\Psi$ can have different modes than the original $q_\Theta$. While this non-invariance and change of geometry are useful, e.g. for normalizing flows [68], they cause issues when invariance is desirable, such as in Bayesian inference.

> To obtain invariance in density maximization, transform the density function under the Riemannian volume form as an unconstrained function. In particular when $\boldsymbol{G} \equiv \boldsymbol{I}$ in $\Theta$, this gives the invariant transformation of a Lebesgue density $q_\Theta$, i.e. $q_\Psi = q_\Theta \circ \varphi^{-1}$.

## 4   Related Work

While reparametrization has been extensively used specifically due to its "non-invariance", e.g. in normalizing flows [63, 68], optimization [19, 69], and Bayesian inference [64, 72], our work is not

---

[2]In the sense that the parallelepiped spanned by an orthonormal basis has volume one.

at odds with them. Instead, it gives further insights into the inner working of those methods: They are formulated by *not* following the geometric rules laid out in Section 2.2, and thus in this case, reparametrization implies a change of metric and hence a change of geometry of the parameter space. They can thus be seen as methods for metric learning [35, 77], i.e. finding the "best" $\boldsymbol{G}$ for the problem at hand, and are compatible with our work since we do not assume a particular metric.

Hessian-based sharpness measures have been extensively used to measure the generalization of neural nets [13, 27, 53]. However, as Dinh et al. [20] pointed out, they are not invariant under reparametrization. Previous work has proposed the Fisher metric to obtain invariant flatness measures [30, 38, 49]. In this work, we have argued that while the Fisher metric is a good choice due to its automatic-invariance property among other statistical benefits [49], any metric is invariant if one follows the proper transformation rules faithfully. That is, the Fisher metric is not necessary if invariance is the only criterion. Similar reasoning about the Fisher metric has been argued in optimization [3, 37, 51, 55, 65, 71] and MAP estimation [21, 32]: the Fisher metric is often used due to its invariance. However, we have discussed that it is not even the unique automatically invariant metric (Appendix C), so the invariance of the Fisher should not be the decisive factor when selecting a metric. By removing invariance as a factor, our work gives practitioners more freedom in selecting a more suitable metric for the problem at hand, beyond the usual Fisher metric.

Finally, the present work is not limited to just the parameter space. For instance, it is desirable for the latent spaces of variational autoencoders [40] to be invariant under reparametrization [2, 25]. The insights of our work can be implemented directly to latent spaces since they are also manifolds [9].

## 5 Some Applications

We present applications in infinite-width Bayesian NNs, model selection, and optimization, to show some directions where the results presented above, and invariance theory in general, can be useful. They are not exhaustive, but we hope they can be an inspiration and foundation for future research.

### 5.1 Infinite-width neural networks

Bayesian NNs tend to Gaussian processes (GPs) as their widths go to infinity [57, 60]. It is widely believed that different parametrizations of an NN yield different limiting kernels [75]. For instance, the *NTK parametrization (NTP)* yields the NTK [29], and the *standard parametrization (SP)* yields the NNGP kernel [45].

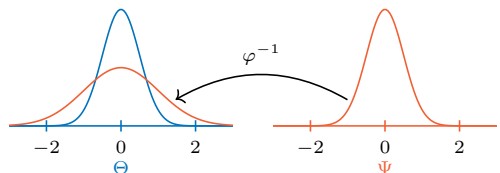

Figure 4: Even though $\mathcal{N}(\boldsymbol{\theta} \mid \boldsymbol{0}, \sigma^2/h\boldsymbol{I})$ in the SP and $\mathcal{N}(\boldsymbol{\psi} \mid \boldsymbol{0}, \sigma^2/h\boldsymbol{I})$ in the NTP, they correspond to different priors. Here, we use $\sigma = 1$, $h = 4$.

Due to their nomenclatures and the choice of priors, it is tempting to treat the NTP and SP as reparametrization of each other. That is, at each layer, the SP parameter $\boldsymbol{\theta}$ transforms into $\boldsymbol{\psi} = \varphi(\boldsymbol{\theta}) = \sigma/\sqrt{h}\boldsymbol{\theta}$ where $h$ is the previous layer's width. This induces the same prior on both $\boldsymbol{\theta}$ and $\boldsymbol{\psi}$, i.e. $\mathcal{N}(\boldsymbol{0}, \sigma^2/h\boldsymbol{I})$, and thus one might guess that invariance should be immediate. However, if they are indeed a reparametrization of the other, the NN $f_{\boldsymbol{\theta}}$ must transform as $f_{\boldsymbol{\psi}} := f_{\boldsymbol{\theta}} \circ \varphi^{-1}(\boldsymbol{\psi})$ and any downstream quantities, including the NTK, must also be invariant. This is a contradiction since the functional form of the NTP shown in Jacot et al. [29] does not equal $f_{\boldsymbol{\psi}}$, and the NTK diverges in the SP but not in the NTP [70]. The SP and NTP are thus not just reparametrizations.

Instead, we argue that the SP and NTP are two completely different choices of the NN's architectures, hyperparameters (e.g. learning rate), and priors—see Fig. 4 for intuition and Appendix E.1 for the details. Seen this way, it is thus not surprising that different "parametrization" yields different limiting behavior. Note that this argument applies to other "parametrizations" [e.g. 15, 70, 75, 76].

Altogether, our work complements previous work and opens up the avenue for constructing non-trivial infinite-width NNs in a "Bayesian" way, in the sense that we argue to achieve a desired limiting behavior by varying the model (i.e. architecture, functional form) and the prior (including over hyperparameters) instead of varying the parametrization. This way, one may leverage Bayesian analysis, e.g. model selection via the marginal likelihood [28, 54], for studying infinite-width NNs.

## 5.2 The Laplace marginal likelihood

The Laplace log-marginal likelihood (LML) of a model $\mathcal{M}$ with parameter in $\mathbb{R}^d$—useful for Bayesian model comparison [11, Sec. 4.4.1]—is defined by [54]

$$\log Z(\mathcal{M}; \boldsymbol{\theta}_{\mathrm{MAP}}) := -\mathcal{L}(\boldsymbol{\theta}_{\mathrm{MAP}}) + \log p(\boldsymbol{\theta}_{\mathrm{MAP}}) - \frac{d}{2}\log(2\pi) + \frac{1}{2}\log\det \boldsymbol{H}(\boldsymbol{\theta}_{\mathrm{MAP}}). \qquad (5)$$

Let $\varphi : \boldsymbol{\theta}_{\mathrm{MAP}} \mapsto \boldsymbol{\psi}_{\mathrm{MAP}}$ be a reparametrization of $\mathbb{R}^d$. It is of interest to answer whether $\log Z$ is invariant under $\varphi$, because if it is not, then $\varphi$ introduces an additional confounder in the model selection and therefore might yield spurious/inconsistent results. For instance, there might exists a reparametrization s.t. $\log \tilde{Z}(\mathcal{M}_1) \geq \log \tilde{Z}(\mathcal{M}_2)$ even though originally $\log Z(\mathcal{M}_1) < \log Z(\mathcal{M}_2)$. The question of "which model is better" thus cannot be answered definitively.

As we have established in Section 3.3, the first and second terms, i.e. $\mathcal{L}_{\mathrm{MAP}}$, are invariant under the Riemannian volume measure. At a glance, the remaining terms do not seem to be invariant due to the transformation of the bilinear-Hessian as discussed in Sec. 3.1. Here, we argue that these terms are invariant when one considers the derivation of the Laplace LML, not just the individual terms in (5).

In the Laplace approximation, the last two terms of $\log Z$ are the normalization constant of the unnormalized density [18] $h(\boldsymbol{\theta}) := \exp\left(-\frac{1}{2}\boldsymbol{d}^\top \boldsymbol{H}(\boldsymbol{\theta}_{\mathrm{MAP}})\boldsymbol{d}\right)$, where $\boldsymbol{d} := (\boldsymbol{\theta} - \boldsymbol{\theta}_{\mathrm{MAP}})$. Notice that $\boldsymbol{d}$ is a tangent vector at $\boldsymbol{\theta}_{\mathrm{MAP}}$ and $\boldsymbol{H}(\boldsymbol{\theta}_{\mathrm{MAP}})$ is a bilinear form acting on $\boldsymbol{d}$. Using the transformation rules in Example 4, it is straightforward to show that $h \mapsto h \circ \varphi^{-1}$—see

Table 1: The Laplace LML $\log Z$ and its decomposition on a NN under different parametrizations. $\mathcal{L}$ and "Rest" stand for the first and the remaining terms in (5).

| Param. | $\log Z$ | $-\mathcal{L}$ | Rest |
|---|---|---|---|
| Cartesian | -212.7±3.4 | -143.4±0.0 | -69.3±3.4 |
| WeightNorm | -227.1±3.6 | -143.4±0.0 | -83.7±3.6 |

Appendix E.3. Since $Z$ is fully determined by $h$, this suggests that $\log Z(\mathcal{M}; \boldsymbol{\theta}_{\mathrm{MAP}})$ also transforms into $\log \tilde{Z}(\mathcal{M}; \varphi^{-1}(\boldsymbol{\psi}_{\mathrm{MAP}}))$, where $\boldsymbol{\psi}_{\mathrm{MAP}} = \varphi(\boldsymbol{\theta}_{\mathrm{MAP}})$. This is nothing but the transformation of a standard, unconstrained function on $\mathbb{R}^d$, thus $\log Z$ is invariant.

We show this numerically in Table 1. We train a network in the Cartesian parametrization and obtain its $\log Z$. Then we reparametrize the net with WeightNorm and naïvely compute $\log Z$ again. These $\log Z$'s are different because the WeightNorm introduces more parameters than the Cartesian one, even though the degrees of freedom are the same. Moreover, the Hessian-determinant is not invariant under autodiff. However, when transformed as argued above, $\log Z$ is trivially invariant.

## 5.3 Biases of preconditioned optimizers

The loss landscape depends on the metric assumed in the parameter space through the Hessian operator $\boldsymbol{E} = \boldsymbol{G}^{-1}\boldsymbol{H}$. The assumption on $\boldsymbol{G}$ in practice depends on the choice of the optimizer. For instance, under the default assumption of the Cartesian coordinates, using GD implies the Euclidean metric $\boldsymbol{G} \equiv \boldsymbol{I}$, and ADAM uses the gradient-2nd-moment metric [39].

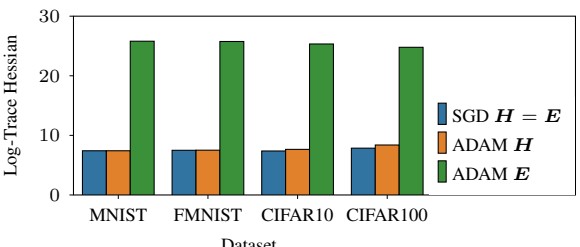

Figure 5: The effect of viewing the Hessian as a linear map $\boldsymbol{E}$ to measure sharpness at minima. ADAM finds much sharper minima when the geometry of the parameter space is taken into account.

We argue that explicitly including the metric is not just theoretically the correct thing to do (since it induces invariance, and by Riemannian geometry), but also practically beneficial. Fig. 5 compares measurements of sharpness of minima (Hessian-trace). Using the metric-aware Hessian operator $\boldsymbol{E}$, one can show definitively (i.e. independent of the choice of parametrization) that ADAM tends to obtain much sharper minima than SGD. The benefits of using the Hessian operator have also been confirmed in previous work. Cohen et al. [16] argue that when analyzing the optimization dynamic of an adaptive/preconditioned GD algorithm, one should take the preconditioner into account when measuring Hessian-based sharpness measures. From this, they demonstrate a sharpness-evolution behavior known in vanilla GD, allowing a comparison between vanilla and adaptive GD.

# 6 Conclusion

In this work, we addressed the invariance and invariance associated with the reparametrization of neural nets. We started with the observation that the parameter space is a Riemannian manifold, albeit often a trivial one. This raises the question of why one should observe non-invariance in neural nets, whereas, by definition, Riemannian manifolds are invariant under a change of coordinates. As we showed, this discrepancy only arises if the transformations used in the construction of a neural net along with an optimizer ignore the implicitly assumed metric. By acknowledging the metric and using the transformation rules associated with geometric objects, invariance and invariance under reparametrization are then guaranteed. Our results provide a geometric solution towards full invariance of neural nets—it is compatible with and complementary to other works that focus on invariance under group-action symmetries, both in the weight and input spaces of neural networks.

## Acknowledgments and Disclosure of Funding

The authors gratefully acknowledge financial support by the European Research Council through ERC StG Action 757275 / PANAMA; the DFG Cluster of Excellence "Machine Learning - New Perspectives for Science", EXC 2064/1, project number 390727645; the German Federal Ministry of Education and Research (BMBF) through the Tübingen AI Center (FKZ: 01IS18039A); the Deutsche Forschungsgemeinschaft (DFG, German Research Foundation) in the frame of the priority programme SPP 2298 "Theoretical Foundations of Deep Learning"—FKZ HE 7114/5-1; and funds from the Ministry of Science, Research and Arts of the State of Baden-Württemberg. Resources used in preparing this research were provided, in part, by the Province of Ontario, the Government of Canada through CIFAR, and companies sponsoring the Vector Institute. AK & FD are grateful to the International Max Planck Research School for Intelligent Systems (IMPRS-IS) for support. AK, FD, and PH are also grateful to Frederik Künstner and Runa Eschenhagen for fruitful discussions.

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

## Appendix A    A Simple Working Example

Let $\Theta := \mathbb{R}^d$ and let $\mathcal{L} : \mathbb{R}^d \to \mathbb{R}$ defined by $\mathcal{L}(\boldsymbol{\theta}) := \sum_{i=1}^d (\boldsymbol{\theta}_i)^2$ be a twice-differentiable function on $\Theta$. Suppose $\varphi : \Theta \to \Psi$ defined by $\varphi(\boldsymbol{\theta}) = 3\boldsymbol{\theta} =: \boldsymbol{\psi}$ be the reparametrization of choice. In this example, we shall see how this often-used reparametrization leads to pathologies in the values derived from the Hessian of $\mathcal{L}$ computed by an autodiff system.

Notice that the Jacobian of $\varphi$ is given by $\boldsymbol{J}(\boldsymbol{\theta}) = \mathrm{diag}(3, \ldots, 3) \in \mathbb{R}^{d \times d}$ and its inverse is $\boldsymbol{J}^{-1}(\boldsymbol{\psi}) = \mathrm{diag}(1/3, \ldots, 1/3)$. The autodiff Hessian matrix is given by $\boldsymbol{H}(\boldsymbol{\theta}) \equiv 2\boldsymbol{I} \in \mathbb{R}^{d \times d}$. When we transform $\boldsymbol{\theta} \to \boldsymbol{\psi}$, the function $\mathcal{L}(\boldsymbol{\theta})$ becomes $\hat{\mathcal{L}}(\boldsymbol{\psi}) := \sum_{i=1}^d (1/3\psi_i)^2$. The autodiff will take into account this change and thus will compute the Hessian $\hat{\boldsymbol{H}}$ of $\hat{\mathcal{L}}$ as

$$\hat{\boldsymbol{H}}_{ij}(\boldsymbol{\psi}) = \mathrm{diag}(1/3, \ldots, 1/3)\,(2\boldsymbol{I})\,\mathrm{diag}(1/3, \ldots, 1/3) = \mathrm{diag}(2/9, \ldots, 2/9).$$

Notice that, $\det \boldsymbol{H}(\boldsymbol{\theta}) \equiv 2^d$, meanwhile, we have $\det \hat{\boldsymbol{H}}(\boldsymbol{\psi}) \equiv (2/9)^d$. Therefore, the Hessian determinant computed by an autodiff system is not invariant under reparametrization.

But if instead we explicitly take into account the metric into the Hessian, which by default is (implicitly) chosen to be $\boldsymbol{G}(\boldsymbol{\theta}) \equiv \boldsymbol{I}$, resulting in the seemingly redundant expression $\boldsymbol{E}(\boldsymbol{\theta}) = \boldsymbol{G}(\boldsymbol{\theta})^{-1}\boldsymbol{H}(\boldsymbol{\theta}) = \boldsymbol{I}^{-1}(2\boldsymbol{I})$, and perform the correct transformation on both identity matrices, we obtain

$$
\begin{aligned}
\hat{\boldsymbol{E}}(\boldsymbol{\psi}) &= (\mathrm{diag}(1/3, \ldots, 1/3)\,\boldsymbol{I}\,\mathrm{diag}(1/3, \ldots, 1/3))^{-1}\mathrm{diag}(1/3, \ldots, 1/3)\,(2\boldsymbol{I})\,\mathrm{diag}(1/3, \ldots, 1/3) \\
&= \mathrm{diag}(3, \ldots, 3)\,\underline{\mathrm{diag}(3, \ldots, 3)\mathrm{diag}(1/3, \ldots, 1/3)}\,(2\boldsymbol{I})\,\mathrm{diag}(1/3, \ldots, 1/3) \\
&= \mathrm{diag}(3, \ldots, 3)\,(2\boldsymbol{I})\,\mathrm{diag}(1/3, \ldots, 1/3).
\end{aligned}
$$

Thus, the determinant of the transformed "metric-conditioned Hessian" equals

$$\det \hat{\boldsymbol{E}}(\boldsymbol{\psi}) = (\det \mathrm{diag}(3, \ldots, 3))\,(\det 2\boldsymbol{I})(\det \mathrm{diag}(1/3, \ldots, 1/3)) = 3^d 2^d (1/3)^d = 2^d,$$

which coincides with the determinant of the original $\boldsymbol{E}$.

## Appendix B    Derivations

**Note.**   Throughout this section, we use the Einstein summation convention: If the same index appears twice, once as an upper index and once as a lower index, we sum them over. For example: $z = x^i y_i$ means $z = \sum_i x^i y_i$ and $B_j^k = A_{ij}^k x^i$ means $B_j^k = \sum_k A_{ij}^k x^i$, etc. Specifically for partial derivatives, the index of the denominator is always treated as a lower index. ∎

### B.1    The Riemannian Hessian Under Reparametrization

Let $\mathcal{L} : M \to \mathbb{R}$ be a function on a Riemannian manifold $M$ with metric $G$. The Riemannian Hessian $\mathrm{Hess}\,\mathcal{L}$ of $\mathcal{L}$ is defined in coordinates $\theta$ by

$$\boldsymbol{H}_{ij} = \frac{\partial^2 \mathcal{L}}{\partial \boldsymbol{\theta}^i \partial \boldsymbol{\theta}^j} - \boldsymbol{\Gamma}_{ij}^k \frac{\partial \mathcal{L}}{\partial \boldsymbol{\theta}^k}, \tag{6}$$

where $\boldsymbol{\Gamma}_{ij}^k$ is the connection coefficient.

Under a change of coordinates $\varphi : \boldsymbol{\theta} \mapsto \boldsymbol{\psi}$, we have $\tilde{\mathcal{L}} = \mathcal{L} \circ \varphi^{-1}$ and

$$\tilde{\boldsymbol{\Gamma}}_{ij}^k = \boldsymbol{\Gamma}_{mn}^o \frac{\partial \boldsymbol{\psi}^k}{\partial \boldsymbol{\theta}^o} \frac{\partial \boldsymbol{\theta}^m}{\partial \boldsymbol{\psi}^i} \frac{\partial \boldsymbol{\theta}^n}{\partial \boldsymbol{\psi}^j} + \frac{\partial^2 \boldsymbol{\theta}^o}{\partial \boldsymbol{\psi}^i \partial \boldsymbol{\psi}^j} \frac{\partial \boldsymbol{\psi}^k}{\partial \boldsymbol{\theta}^o}, \tag{7}$$

where $m$, $n$, $o$ are just dummy indices—present to express summations. Note that the transformation rule for $\boldsymbol{\Gamma}_{ij}^k$ implies that it is *not* a tensor—to be a tensor, there must not be the second term in the previous formula.

Using (7) and the standard chain & product rules to transform the partial derivatives in (6), we obtain the coordinate representation of the transformed Hessian Hess $\tilde{\mathcal{L}}$:

$$
\begin{aligned}
\tilde{H}_{ij} &= \frac{\partial^2(\mathcal{L} \circ \varphi^{-1})}{\partial \psi^i \partial \psi^j} - \tilde{\Gamma}^k_{ij} \frac{\partial(\mathcal{L} \circ \varphi^{-1})}{\partial \psi^k} \\
&= \frac{\partial^2 \mathcal{L}}{\partial \theta^m \partial \theta^n} \frac{\partial \theta^m}{\partial \psi^i} \frac{\partial \theta^n}{\partial \psi^j} + \frac{\partial \mathcal{L}}{\partial \theta^o} \frac{\partial^2 \theta^o}{\partial \psi^i \partial \psi^j} - \left( \Gamma^o_{mn} \frac{\partial \psi^k}{\partial \theta^o} \frac{\partial \theta^m}{\partial \psi^i} \frac{\partial \theta^n}{\partial \psi^j} + \frac{\partial^2 \theta^o}{\partial \psi^i \partial \psi^j} \frac{\partial \psi^k}{\partial \theta^o} \right) \frac{\partial \mathcal{L}}{\partial \theta^o} \frac{\partial \theta^o}{\partial \psi^k} \\
&= \frac{\partial^2 \mathcal{L}}{\partial \theta^m \partial \theta^n} \frac{\partial \theta^m}{\partial \psi^i} \frac{\partial \theta^n}{\partial \psi^j} + \frac{\partial \mathcal{L}}{\partial \theta^o} \frac{\partial^2 \theta^o}{\partial \psi^i \partial \psi^j} - \Gamma^o_{mn} \frac{\partial \psi^{\cancel{k}}}{\partial \theta^o} \frac{\partial \theta^m}{\partial \psi^i} \frac{\partial \theta^n}{\partial \psi^j} \frac{\partial \mathcal{L}}{\partial \theta^o} \frac{\partial \theta^{\cancel{o}}}{\partial \psi^k} - \frac{\partial^2 \theta^o}{\partial \psi^i \partial \psi^j} \frac{\partial \psi^{\cancel{k}}}{\partial \theta^o} \frac{\partial \mathcal{L}}{\partial \theta^o} \frac{\partial \theta^{\cancel{o}}}{\partial \psi^k} \\
&= \frac{\partial^2 \mathcal{L}}{\partial \theta^m \partial \theta^n} \frac{\partial \theta^m}{\partial \psi^i} \frac{\partial \theta^n}{\partial \psi^j} + \cancel{\frac{\partial \mathcal{L}}{\partial \theta^o} \frac{\partial^2 \theta^o}{\partial \psi^i \partial \psi^j}} - \Gamma^o_{mn} \frac{\partial \theta^m}{\partial \psi^i} \frac{\partial \theta^n}{\partial \psi^j} \frac{\partial \mathcal{L}}{\partial \theta^o} - \cancel{\frac{\partial^2 \theta^o}{\partial \psi^i \partial \psi^j} \frac{\partial \mathcal{L}}{\partial \theta^o}} \\
&= \frac{\partial \theta^m}{\partial \psi^i} \frac{\partial \theta^n}{\partial \psi^j} \left( \frac{\partial^2 \mathcal{L}}{\partial \theta^m \partial \theta^n} - \Gamma^o_{mn} \frac{\partial \mathcal{L}}{\partial \theta^o} \right) \\
&= \frac{\partial \theta^m}{\partial \psi^i} \frac{\partial \theta^n}{\partial \psi^j} H_{mn}.
\end{aligned}
$$

(8)

In the matrix form, we can write the above as $\tilde{H} = J^{-\top} H J^{-1}$, where $J$ is the Jacobian of $\varphi$. Thus, the Riemannian Hessian at any $\theta$ (not just at critical points) transforms just like the metric and thus invariant as discussed in Example 4. Note: this only holds when the term containing the connection coefficients $\Gamma^k_{ij}$ is explicitly considered. In particular, the Euclidean Hessian does not follow this tensorial transformation under autodiff due to the fact that (i) $\Gamma^k_{ij} = 0$ for any $i$, $j$, $k$ and thus dropped from the equation, *and* (ii) autodiff is not designed to handle advanced geometric objects like $\Gamma^k_{ij}$.

## B.2 Hessian-Trace Under Reparametrization

Let $\mathcal{L} : \mathbb{R}^d \to \mathbb{R}$ be a function of $\mathbb{R}^d$ under the Cartesian coordinates and $G$ a Riemannian metric. The ***Riemannian trace*** of the Hessian matrix $H$ of $\mathcal{L}$ is defined by [47]:

$$
(\text{tr}_G H)(\theta) = \text{tr}(G(\theta)^{-1} H(\theta)). \tag{9}
$$

That is, it is defined as the standard trace of the Hessian operator $E$.

Let $\varphi : \theta \mapsto \psi$ be a reparametrization on $\mathbb{R}^d$. Then, using (2) and the property $\text{tr}(AB) = \text{tr}(BA)$ twice, the Riemannian trace of the Hessian transforms into

$$
\begin{aligned}
(\text{tr}_{\tilde{G}} \tilde{H})(\psi) &= \text{tr}(\tilde{E}(\psi)) \\
&= \text{tr}((J^{-1}(\psi))^{-1} G(\varphi^{-1}(\psi)) H(\varphi^{-1}(\psi)) J^{-1}(\psi)) \\
&= \text{tr}(G(\varphi^{-1}(\psi)) H(\varphi^{-1}(\psi))) \\
&= (\text{tr}_G H)(\varphi^{-1}(\psi)).
\end{aligned} \tag{10}
$$

Since $\psi = \varphi(\theta)$, we have that $(\text{tr}_{\tilde{G}} \tilde{H})(\psi) = (\text{tr}_G H)(\theta)$ for any given $\theta$. Therefore, the trace of the Hessian operator (or the Riemannian trace of the Hessian) is invariant.

## B.3 Hessian-Eigenvalues Under Reparametrization

We use the setting from the preceding section. Recall that $\lambda$ is an eigenvalue of the linear map $E(\theta) = G(\theta)^{-1} H(\theta)$ on the tangent space at $z \in \mathbb{R}^d$ that is represented $\theta$ if $E(\theta)v = \lambda v$ for an eigenvector $v \in T_z \mathbb{R}^d$. We shall show that $\tilde{\lambda}$, the eigenvalue under under the reparametrization $\varphi : \theta \mapsto \psi$, equals the original eigenvalue $\lambda$.

Using the transformation rule of $E(\theta)$ in (2) and the transformation rule of tangent vectors in (4), along with the relation $(J^{-1}(\psi))^{-1} = J(\varphi^{-1}(\psi))$, we get

$$
\begin{aligned}
\tilde{E}(\psi)\tilde{v} &= \tilde{\lambda}\tilde{v} \\
(J^{-1}(\psi))^{-1} G(\varphi^{-1}(\psi)) H(\varphi^{-1}(\psi)) \cancel{J^{-1}(\psi) J(\varphi^{-1}(\psi))} v &= \tilde{\lambda} J(\varphi^{-1}(\psi)) v \\
J(\varphi^{-1}(\psi)) G(\varphi^{-1}(\psi)) H(\varphi^{-1}(\psi)) v &= \tilde{\lambda} J(\varphi^{-1}(\psi)) v \\
\cancel{(J(\varphi^{-1}(\psi)))^{-1} J(\varphi^{-1}(\psi))} G(\varphi^{-1}(\psi)) H(\varphi^{-1}(\psi)) v &= \tilde{\lambda} v \\
E(\varphi^{-1}(\psi)) v &= \tilde{\lambda} v.
\end{aligned} \tag{11}
$$

Since $\varphi^{-1}(\boldsymbol{\psi}) = \boldsymbol{\theta}$, we arrive back at the original equation before the reparametrization and thus we conclude that $\tilde{\lambda} = \lambda$.

## Appendix C  The Invariance of the Fisher Metric

We have seen that any metric on the parameter space yields inv(equi)variance when all transformation rules of geometric objects are followed. However, in practical settings, one uses autodiff libraries [1, 66], whose job is to compute only the elementary rules of derivatives such as the chain and product rules. Notice that derivatives and Hessians (at least at critical points) are transformed correctly under autodiff, while the metric is not in general. It is thus practically interesting to find a family of metrics that transform correctly and automatically under reparametrization, given only an autodiff library. Under those metrics, the aforementioned inv(equi)variances can thus be obtained effortlessly.

Martens [55, Sec. 12] mentions the following family of curvature matrices satisfying this transformation behavior

$$ \boldsymbol{B}(\boldsymbol{\theta}) \propto \mathbb{E}_{\boldsymbol{x},\boldsymbol{y}\sim D} \left[ \boldsymbol{J}(\boldsymbol{\theta};\boldsymbol{x})^\top \boldsymbol{A}(\boldsymbol{\theta},\boldsymbol{x},\boldsymbol{y}) \boldsymbol{J}(\boldsymbol{\theta};\boldsymbol{x}) \right] , \tag{12} $$

where $\boldsymbol{J}(\boldsymbol{\theta};\cdot)$ is the network's Jacobian $\partial f(\,\cdot\,;\boldsymbol{\theta})/\partial\boldsymbol{\theta}$, with an arbitrary data distribution $D$ and an invertible matrix $\boldsymbol{A}$ that transforms like $\boldsymbol{A}(\varphi^{-1}(\boldsymbol{\psi}))$ under $\varphi : \boldsymbol{\theta} \mapsto \boldsymbol{\psi}$. Under a reparametrization $\varphi$, an autodiff library will compute

$$ \hat{\boldsymbol{B}}(\boldsymbol{\psi}) = \boldsymbol{J}^{-1}(\boldsymbol{\psi})^\top \boldsymbol{B}(\varphi^{-1}(\boldsymbol{\psi})) \boldsymbol{J}^{-1}(\boldsymbol{\psi}) , \tag{13} $$

given $\mathcal{L} \circ \varphi^{-1}$, due to the elementary transformation rule of $\boldsymbol{J}(\boldsymbol{\theta};\cdot)$, just like Example 4(c). This family includes the Fisher and the generalized Gauss-Newton matrices, as well as the empirical Fisher matrix [44, 55].

However, note that any $\boldsymbol{B}$ as above is sufficient. Thus, this family is much larger than the Fisher metric, indicating that automatic invariance is not unique to that metric and its denominations. Moreover, in practice, these metrics are often substituted by structural approximations, such as their diagonal and Kronecker factorization [56]. This restricts the kinds of reparametrizations under which these approximate metrics transform automatically: Take the diagonal of (12) as an example. Its $\psi$-representation computed by the autodiff library will only match the correct transformation for element-wise reparametrizations, whose Jacobian $\boldsymbol{J}(\boldsymbol{\theta})$ is diagonal. On top of that, in practical algorithms, such approximations are usually combined with additional techniques such as damping or momentum, which further break their automaticness.

Due to the above and because any metric is theoretically in(equi)variant if manual intervention is performed, the inv(equi)variance property of the Fisher metric should not be the determining factor of using the Fisher metric. Instead, one should look into its more unique properties such as its statistical efficiency [3] and guarantees in optimization [34, 78]. Automatic transformation is but a cherry on top.

It is interesting for future work to extend autodiff libraries to take into account the invariant transformation rules of geometric objects. By doing so, *any* metric—not just the Fisher metric—will yield invariance *automatically*.

## Appendix D  General Manifold: Local Coordinate Charts

In the main text, we have exclusively used global coordinate charts $(\mathbb{R}^d, \theta)$ and $(\mathbb{R}^d, \psi)$—these charts cover the entire $\mathbb{R}^d$ and $\theta$, $\psi$ are homeomorphisms on $\mathbb{R}^d$. However, there exist other coordinate systems that are not global, i.e. they are constructed using multiple charts $\{(U_i \subseteq \mathbb{R}^d, \theta_i : U_i \to \Theta_i \subseteq \mathbb{R}^d)\}_i$ s.t. $\mathbb{R}^d = \cup_i U_i$ and each $\theta_i$ is a diffeomorphism on $U_i$.

If $\{(U_i, \theta_i)\}_i$ and $\{(V_j, \psi_j)\}_j$ be two local coordinate systems of $\mathbb{R}^d$. Let $(U_i, \theta_i)$ and $(V_j, \psi_j)$ be an arbitrary pair of charts from the above collections where $U_i \cap V_j \neq \varnothing$. In this setting, reparametrization between these two charts amounts to the condition that the transition map $\varphi$ between $\theta(U_i \cap V_j)$ and $\psi(U_i \cap V_j)$ is a diffeomorphism, see Fig. 6. Reparametrization on the whole manifold can then be defined if for all pairs of two charts from two coordinate systems, their transition maps are all diffeomorphism whenever they overlap with each other.

Note that this definition is a generalization of global coordinate charts. In this case, there is only a single chart for each coordinate system, i.e. $(U, \theta)$ and $V, \psi$, and they trivially overlap since $U = \mathbb{R}^d$ and $V = \mathbb{R}^d$. Moreover, there is only a single diffeomorphism of concern, as shown in Fig. 3.

Our results in the main text hold in this general case by simply replacing the domain for the discussion to be the overlapping regions of a pair of two charts, instead of the whole $\mathbb{R}^d$.

Figure 6: The diffeomorphism $\varphi$ is defined from $\theta(U \cap V)$ to $\psi(U \cap V)$.

## Appendix E  Details on Applications

### E.1  Infinite-Width Neural Networks

**Note**   We use a 2-layer NN without bias for clarity. The extension to deep networks follows directly by induction—see e.g. [6, 45]. We also use the same prior variance $\sigma^2$ without loss of generalization for simplicity. See also Kristiadi et al. [41, Appendix A] for further intuition. Recall the property of Gaussians under linear transformation: $\boldsymbol{z} \sim \mathcal{N}(\boldsymbol{\mu}, \boldsymbol{\Sigma}) \implies \boldsymbol{Az} \sim \mathcal{N}(\boldsymbol{A\mu}, \boldsymbol{A\Sigma A}^\top)$.

**Neural-network Gaussian process (NNGP)**

Let $f(\boldsymbol{x}) : \mathbb{R}^d \to \mathbb{R}$ defined by $f(\boldsymbol{x}) := \boldsymbol{w}^\top \phi(\boldsymbol{Wx})$ be a real-valued, 2-layer NN with parameter $\boldsymbol{\theta} := \{\boldsymbol{W} \in \mathbb{R}^{h \times n}, \boldsymbol{w} \in \mathbb{R}^h\}$ and component-wise nonlinearity $\phi$. Note that we assume $\boldsymbol{x}$ is i.i.d. Let $\text{vec}(\boldsymbol{W}) \sim \mathcal{N}(\boldsymbol{0}, \sigma^2 \boldsymbol{I})$ and $\boldsymbol{w} \sim \mathcal{N}(\boldsymbol{0}, \frac{\sigma^2}{h}\boldsymbol{I})$ be priors over the weights. This parametrization of the weights and the priors is called the ***standard parametrization (SP)*** [45, 60].

**Step (a)**   Given a particular preactivation value $\boldsymbol{z}$, we have a linear model $f(\boldsymbol{x}) = \boldsymbol{w}^\top \phi(\boldsymbol{z})$. We can view this as a Gaussian process with mean and covariance

$$\mathbb{E}[f(\boldsymbol{x})] = \boldsymbol{0}^\top \phi(\boldsymbol{z}) = 0,$$

$$\text{Cov}[f(\boldsymbol{x}), f(\boldsymbol{x}')] = \frac{\sigma^2}{h} \phi(\boldsymbol{z})^\top \phi(\boldsymbol{z}') = \frac{\sigma^2}{h} \sum_{i=1}^{h} \phi(\boldsymbol{z}_i)\, \phi(\boldsymbol{z}_i').$$

Taking the limit as $h \to \infty$, the mean stays trivially zero and we have by the law of large numbers:

$$K(\boldsymbol{x}, \boldsymbol{x}') := \lim_{h \to \infty} \text{Cov}[f(\boldsymbol{x}), f(\boldsymbol{x}')] = \lim_{h \to \infty} \frac{\sigma^2}{h} \sum_{i=1}^{h} \phi(\boldsymbol{z}_i)\, \phi(\boldsymbol{z}_i') = \sigma^2 \mathop{\mathbb{E}}_{\boldsymbol{z}_i, \boldsymbol{z}_i'} [\phi(\boldsymbol{z}_i)\phi(\boldsymbol{z}_i')].$$

In particular, both the mean and covariance over the output does *not* depend on the particular realization of the hidden units $\phi(\boldsymbol{z})$. That is, they only depend on the *distribution of $\boldsymbol{z}$ induced by the prior*, which we will obtain now.

**Step (b)**   Notice that each $\boldsymbol{z}_i = \boldsymbol{W}_i^\top \boldsymbol{x}$ where $\boldsymbol{W}_i$ is the $i$-th row of $\boldsymbol{W}$. This is a linear model and thus, $\boldsymbol{z}$ is distributed as a GP with mean and covariance

$$\mathbb{E}[\boldsymbol{z}_i] = \boldsymbol{0}^\top \boldsymbol{x} = 0,$$

$$\text{Cov}[\boldsymbol{z}_i, \boldsymbol{z}_i'] = \sigma^2 \boldsymbol{x}^\top \boldsymbol{x}' =: K_z(\boldsymbol{z}_i, \boldsymbol{z}_i').$$

So, $\boldsymbol{z}_i \sim \mathcal{GP}(0, K_z)$ Since the prior over $\boldsymbol{W}$ is i.i.d., this holds for all $i = 1, \ldots, h$. We can thus now compute the expectation in $K(\boldsymbol{x}, \boldsymbol{x}')$: it is done w.r.t. this GP over $\boldsymbol{z}_i$.

To obtain the GP over the function output of a deep network, simply apply steps (a) and (b) above recursively. The crucial message from this derivation is that as the width of each layer of a deep net goes to infinity, the network loses representation power—the output of each layer only depends on the prior, and not on particular values (e.g. learned) of the previous hidden units. In this sense, an infinite-width $L$-layer NN is simply a linear model with a constant feature extractor induced by the network's first $L - 1$ layers that are fixed at initialization. Note that the kernel $K$ over the function output is called the ***NNGP kernel*** [45].

**Neural tangent kernel (NTK)**

Let us transform $\boldsymbol{w}$ into $\boldsymbol{v} := \frac{\sigma}{\sqrt{h}}\boldsymbol{w}$ and $\boldsymbol{W}$ into $\boldsymbol{V} := \frac{\sigma}{\sqrt{h}}\boldsymbol{W}$ and define the prior to be $\boldsymbol{w} \sim \mathcal{N}(\boldsymbol{0}, \boldsymbol{I})$ and $\mathrm{vec}(\boldsymbol{W}) \sim \mathcal{N}(\boldsymbol{0}, \boldsymbol{I})$. Then, we define the transformed network as $\hat{f}(\boldsymbol{x}) := \boldsymbol{v}^\top \phi(\boldsymbol{V}\boldsymbol{x}) = \frac{\sigma}{\sqrt{h}}\boldsymbol{w}^\top\phi\left(\frac{\sigma}{\sqrt{h}}\boldsymbol{W}\boldsymbol{x}\right)$ with parameter $\boldsymbol{\psi} := \{\boldsymbol{V} \in \mathbb{R}^{h\times n}, \boldsymbol{v} \in \mathbb{R}^h\}$. This is called the *NTK parametrization (NTP)* [29]. We will see below that even though $\boldsymbol{v}$, $\boldsymbol{V}$ have the same prior distributions as $\boldsymbol{w}$, $\boldsymbol{W}$ in the SP, they have different behavior in terms of the NTK.

As before, let us assume a particular preactivation value $\boldsymbol{z}$. The *empirical NTK* (i.e. finite-width NTK) on the last layer is defined by:

$$\hat{\mathcal{K}}(\boldsymbol{x}, \boldsymbol{x}') := \langle \nabla_{\boldsymbol{w}}\hat{f}(\boldsymbol{x}), \nabla_{\boldsymbol{w}}\hat{f}(\boldsymbol{x}')\rangle = \frac{\sigma^2}{h}\sum_{i=1}^h \phi(\boldsymbol{z}_i)\,\phi(\boldsymbol{z}_i').$$

The *(asymptotic) NTK* is obtained by taking the limit of $h \to \infty$:

$$\mathcal{K}(\boldsymbol{x}, \boldsymbol{x}') := \lim_{h\to\infty}\hat{\mathcal{K}}(\boldsymbol{x}, \boldsymbol{x}') = \frac{\sigma^2}{h}\sum_{i=1}^h \phi(\boldsymbol{z}_i)\,\phi(\boldsymbol{z}_i') = \mathop{\mathbb{E}}_{\boldsymbol{z}_i,\boldsymbol{z}_i'}\left[\phi(\boldsymbol{z}_i)\phi(\boldsymbol{z}_i')\right], \tag{14}$$

which coincides the NNGP kernel $K$. Crucially, this is obtained via a backward propagation from the output of the network and thus the linear-Gaussian property we have used to derive the NNGP via forward propagation does not apply.[3] This is why the scaling of $\frac{\sigma}{h}$ is required in the NTP. That is, using the SP, the empirical NTK is not scaled by $\frac{\sigma^2}{h}$ and thus when taking the limit to obtain $\mathcal{K}$, the sum diverges and the limit does not exist.

**Is the NTP a reparametrization of the SP?**

It is tempting to treat the NTP as a reparametrization of the SP—in fact, it is standard in the literature to treat them as two different parametrizations of the same network. However, we show that geometrically, this is inaccurate. Indeed from the geometric perspective, if two functions are reparametrization of each other, they should be invariant, as we have discussed in the main text. Instead, we show that the different limiting behaviors are present because the NTP and SP assume two different functions and two different priors—they are *not* connected by a reparametrization. This clears up confusion and provides a foundation for future work in this field: To obtain a desired limiting behavior, study the network architecture and its prior, instead of the parametrization.

Suppose $\boldsymbol{\psi}$ in the NTP is a reparametrization of $\boldsymbol{\theta}$ in the SP. Then the function $\varphi : \boldsymbol{\theta} \mapsto \boldsymbol{\psi}$ defined by $\boldsymbol{\theta} \mapsto \frac{\sigma}{\sqrt{h}}\boldsymbol{\theta}$ is obviously the smooth reparametrization with an invertible (diagonal) Jacobian $\boldsymbol{J}(\boldsymbol{\theta}) = \frac{\sigma}{\sqrt{h}}\boldsymbol{I}$. In this case, the network in the NTP must be defined by $\tilde{f} = f \circ \varphi^{-1}$, where $f$ is the SP-network, by Example 4. That is, with some abuse of notation,

$$\tilde{f}(\boldsymbol{x}) = \varphi^{-1}(\boldsymbol{v})^\top \phi(\varphi^{-1}(\boldsymbol{V})\boldsymbol{x}) = \boldsymbol{w}^\top\phi(\boldsymbol{W}\boldsymbol{x}) = f(\boldsymbol{x}).$$

This is different from the definition of the NTP-network $\hat{f}(\boldsymbol{x}) = \boldsymbol{v}^\top\phi(\boldsymbol{V}\boldsymbol{x})$. So, obviously, the NTP is not the reparametrization of the SP. Therefore, a clearer way of thinking about the NTP and SP is to treat them as two separate network functions (i.e. two separate architectures)—the scaling factor $\frac{\sigma}{\sqrt{h}}$ should be thought of as part of the layer's functional form instead of as part of the parameter. In particular, they are *not* two representations of a single abstract function.

To verify this, let us compute the NTK of $\tilde{f}(\boldsymbol{x})$ (i.e. treating the scaling as a reparametrization) at its last layer. The derivation is based on Section 3.2. First, notice that the differential $\nabla_{\boldsymbol{w}}f(\boldsymbol{x})$ transforms into $\boldsymbol{J}^{-1}(\boldsymbol{v})^\top\boldsymbol{\nabla}\tilde{f}(\boldsymbol{x})|_{\varphi^{-1}(\boldsymbol{v})}$ for any $\boldsymbol{x} \in \mathbb{R}^n$. Next, notice that the Euclidean metric transforms into $\tilde{\boldsymbol{G}}(\boldsymbol{v}) := \boldsymbol{J}^{-1}(\boldsymbol{v})^\top\boldsymbol{J}^{-1}(\boldsymbol{v})$. So the gradient transforms into $\boldsymbol{J}(\varphi^{-1}(\boldsymbol{v}))\boldsymbol{\nabla}f(\boldsymbol{x})|_{\varphi^{-1}(\boldsymbol{v})}$.[4] Therefore,

---

[3]It still applies for obtaining the distribution of $\boldsymbol{z}_i$. The NTK can thus be thought of as a kernel that arises from performing forward *and* backward propagations once at initialization [6, 75]. This can be seen in the expression of the NTKs on lower layers which decompose into the NNGP and an expression similar to (14), but involving the derivative of $\phi$ [29].

[4]We use the gradient to get the NTK since otherwise it does not make sense to take the inner product of differentials w.r.t. the metric.

Table 2: Test accuracies, averaged over 5 random seeds.

| Methods | MNIST | FMNIST | CIFAR10 | CIFAR100 |
|---------|-------|--------|---------|----------|
| SGD | 99.3 | 92.9 | 94.9 | 76.8 |
| ADAM | 99.2 | 92.6 | 92.4 | 71.9 |

the empirical NTK kernel $\hat{\mathcal{K}}_\Psi$ for $\tilde{f}$ is given by

$$
\begin{aligned}
\hat{\mathcal{K}}_\Psi(\boldsymbol{x}, \boldsymbol{x}') &= \langle \boldsymbol{J}(\varphi^{-1}(\boldsymbol{v}))\boldsymbol{\nabla} f(\boldsymbol{x})|_{\varphi^{-1}(\boldsymbol{v})}, \boldsymbol{J}(\varphi^{-1}(\boldsymbol{v}))\boldsymbol{\nabla} f(\boldsymbol{x}')|_{\varphi^{-1}(\boldsymbol{v})} \rangle_{\tilde{\boldsymbol{G}}(\boldsymbol{v})} \\
&= (\boldsymbol{J}(\varphi^{-1}(\boldsymbol{v}))\boldsymbol{\nabla} f(\boldsymbol{x})|_{\varphi^{-1}(\boldsymbol{v})})^\top \tilde{\boldsymbol{G}}(\boldsymbol{v}) \boldsymbol{J}(\varphi^{-1}(\boldsymbol{v}))\boldsymbol{\nabla} f(\boldsymbol{x}')|_{\varphi^{-1}(\boldsymbol{v})} \\
&= (\boldsymbol{\nabla} f(\boldsymbol{x})|_{\varphi^{-1}(\boldsymbol{v})})^\top \boldsymbol{J}(\varphi^{-1}(\boldsymbol{v}))^\top \boldsymbol{J}^{-1}(\boldsymbol{v})^\top \boldsymbol{J}^{-1}(\boldsymbol{v}) \boldsymbol{J}(\varphi^{-1}(\boldsymbol{v}))\boldsymbol{\nabla} f(\boldsymbol{x}')|_{\varphi^{-1}(\boldsymbol{v})} \\
&= \langle \boldsymbol{\nabla} f(\boldsymbol{x})|_{\varphi^{-1}(\boldsymbol{v})}, \boldsymbol{\nabla} f(\boldsymbol{x}')|_{\varphi^{-1}(\boldsymbol{v})} \rangle.
\end{aligned}
$$

Thus, the empirical NTK is invariant and the asymptotic NTK also is. Therefore, we still have a problem with the NTK blow-up in this parametrization. This reinforces the fact that the difference between the SP and NTP is *not* because of parametrization.

Additionally, let us now inspect the priors in the SP and NTP. In the SP, the prior is $\mathcal{N}(\boldsymbol{\theta} \mid \boldsymbol{0}, \sigma^2/h\boldsymbol{I})$. Therefore, so that we have the same prior in both $\Theta$ and $\Psi$, the prior of $\boldsymbol{\psi} = \varphi(\boldsymbol{\theta})$ must be $\mathcal{N}(\boldsymbol{\psi} \mid \boldsymbol{0}, \boldsymbol{I})$. This is obviously not the case since we have $\mathcal{N}(\boldsymbol{\psi} \mid \boldsymbol{0}, \sigma^2/h\boldsymbol{I})$ because the NTP explicitly defines $\mathcal{N}(\boldsymbol{\theta} \mid \boldsymbol{0}, \boldsymbol{I})$ as the prior of $\boldsymbol{\theta}$. Thus, not only that the SP and NTP assume two different architectures, but they also assume two different prior altogether. It is thus not surprising that the distribution over their network outputs $f(\boldsymbol{x})$, $\hat{f}(\boldsymbol{x})$ are different, both in the finite- and infinite-width regimes.

**Implication**  In his seminal work, Neal [60] concluded that the fact that infinite-width NNs are Gaussian processes disappointing. However, as we have seen in the discussion above, different functional forms, architectures, and priors of NNs yield different limiting behaviors. Therefore, this gives us hope that meaningful, non-GP infinite-width NNs can be obtained. Indeed, Yang and Hu [75], Yang et al. [76] have recently shown us a way to do so. However, they argue that their feature-learning limiting behavior is due to a different parametrization, contrary to the present work. Our work thus complements theirs and opens up the avenue for constructing non-trivial infinite-width NNs in a "Bayesian" way, in the sense that we achieve the desired limiting behaviors by varying the model and the prior.

### E.2  Biases of Preconditioned Optimizers

For MNIST and FMNIST, the network is LeNet. Meanwhile, we use the WiderResNet-16-4 model for CIFAR-10 and -100. For ADAM, we use the default setting suggested by Kingma and Ba [39]. For SGD, we use the commonly-used learning rate of 0.1 with Nesterov momentum 0.9 [26]. The cosine annealing method is used to schedule the learning rate for 100 epochs. The test accuracies are in Table 2. Additionally, in Table 3, we discuss the effect of reparametrization to sharpness on ADAM and SGD.

### E.3  Laplace Marginal Likelihood

Let $\boldsymbol{\theta}_{\mathrm{MAP}}$ be a MAP estimate in an arbitrary $\theta$-coordinates of $\mathbb{R}^d$, obtained by minimizing the MAP loss $\mathcal{L}_{\mathrm{MAP}}$. Let $\log h = -\mathcal{L}_{\mathrm{MAP}}$—note that $\mathcal{L}_{\mathrm{MAP}}$ itself is a log-density function. The Laplace marginal likelihood [18, 28, 54] is obtained by performing a second-order Taylor's expansion:

$$
\log h(\boldsymbol{\theta}) \approx \log h(\boldsymbol{\theta}_{\mathrm{MAP}}) - \frac{1}{2}(\boldsymbol{\theta} - \boldsymbol{\theta}_{\mathrm{MAP}})^\top \boldsymbol{H}(\boldsymbol{\theta}_{\mathrm{MAP}})(\boldsymbol{\theta} - \boldsymbol{\theta}_{\mathrm{MAP}}),
$$

where $\boldsymbol{H}(\boldsymbol{\theta}_{\mathrm{MAP}})$ is the Hessian matrix of $\mathcal{L}_{\mathrm{MAP}}$ at $\boldsymbol{\theta}_{\mathrm{MAP}}$. Then, by exponentiating and taking the integral over $\boldsymbol{\theta}$, we have

$$
Z(\boldsymbol{\theta}_{\mathrm{MAP}}) \approx h(\boldsymbol{\theta}_{\mathrm{MAP}}) \int_{\mathbb{R}^d} \exp\left(-\frac{1}{2}(\boldsymbol{\theta} - \boldsymbol{\theta}_{\mathrm{MAP}})^\top \boldsymbol{H}(\boldsymbol{\theta}_{\mathrm{MAP}})(\boldsymbol{\theta} - \boldsymbol{\theta}_{\mathrm{MAP}})\right) d\boldsymbol{\theta}. \tag{15}
$$

Table 3: Hessian-based sharpness measures can change under reparametrization without affecting the model's generalization (results on CIFAR-10). The generalization gap is the test accuracy, subtracted from the train accuracy—lower is better. Under the default parametrization, SGD achieves lower sharpness and generalizes better than ADAM which achieves higher sharpness. However, one can reparametrize SGD's minimum s.t. it achieves much higher (or lower) sharpness than ADAM while retaining the same generalization performance. Hence, it is hard to study the correlation between sharpness and generalization. This highlights the need for invariance.

| Optimizer | Reparametrization $\boldsymbol{\psi}_{\text{MAP}} = \varphi(\boldsymbol{\theta}_{\text{MAP}})$ | Generalization gap [%] | Sharpness $\text{tr}(\hat{\boldsymbol{H}}(\boldsymbol{\psi}_{\text{MAP}}))$ |
|---|---|---|---|
| ADAM | $\boldsymbol{\psi}_{\text{MAP}} = \boldsymbol{\theta}_{\text{MAP}}$ | $7.2 \pm 0.2$ | $1929.8 \pm 61.2$ |
| SGD | $\boldsymbol{\psi}_{\text{MAP}} = \boldsymbol{\theta}_{\text{MAP}}$ 
 $\boldsymbol{\psi}_{\text{MAP}} = \frac{1}{2}\boldsymbol{\theta}_{\text{MAP}}$ 
 $\boldsymbol{\psi}_{\text{MAP}} = 2\boldsymbol{\theta}_{\text{MAP}}$ | $5.2 \pm 0.2$ | $1531.8 \pm 14.2$ 
 $6143.7 \pm 60.8$ 
 $383.6 \pm 3.3$ |

Since the integral is the normalization constant of the Gaussian $\mathcal{N}(\boldsymbol{\theta} \mid \boldsymbol{\theta}_{\text{MAP}}, \boldsymbol{H}(\boldsymbol{\theta}_{\text{MAP}}))$, we obtain the Laplace log-marginal likelihood (LML):

$$\log Z(\boldsymbol{\theta}_{\text{MAP}}) = -\mathcal{L}_{\text{MAP}}(\boldsymbol{\theta}_{\text{MAP}}) - \frac{d}{2}\log(2\pi) + \log\det\boldsymbol{H}(\boldsymbol{\theta}_{\text{MAP}}).$$

Notice that $\boldsymbol{H}(\boldsymbol{\theta})$ is a bilinear form, acting on the tangent vector $\boldsymbol{d}(\boldsymbol{\theta}_{\text{MAP}}) := (\boldsymbol{\theta} - \boldsymbol{\theta}_{\text{MAP}})$. Under a reparametrization $\varphi : \boldsymbol{\theta} \mapsto \boldsymbol{\psi}$ with $\boldsymbol{\psi}_{\text{MAP}} = \varphi(\boldsymbol{\theta}_{\text{MAP}})$, the term inside the exponent in (15) transforms into

$$-\frac{1}{2}\big(\boldsymbol{J}(\varphi^{-1}(\boldsymbol{\psi}_{\text{MAP}}))\boldsymbol{d}(\varphi^{-1}(\boldsymbol{\psi}_{\text{MAP}}))\big)^{\top}\big(\boldsymbol{J}^{-1}(\boldsymbol{\psi}_{\text{MAP}})^{\top}\boldsymbol{H}(\varphi^{-1}(\boldsymbol{\psi}_{\text{MAP}}))\boldsymbol{J}^{-1}(\boldsymbol{\psi}_{\text{MAP}}))$$
$$\boldsymbol{J}(\varphi^{-1}(\boldsymbol{\psi}_{\text{MAP}}))\boldsymbol{d}(\varphi^{-1}(\boldsymbol{\psi}_{\text{MAP}})),$$

due to the transformations of the tangent vector and the bilinear-Hessian. This simplifies into

$$\exp\left(-\frac{1}{2}\boldsymbol{d}(\varphi^{-1}(\boldsymbol{\psi}_{\text{MAP}}))^{\top}\boldsymbol{H}(\varphi^{-1}(\boldsymbol{\psi}_{\text{MAP}}))\boldsymbol{d}(\varphi^{-1}(\boldsymbol{\psi}_{\text{MAP}}))\right)$$

which always equals the original integrand in (15). Thus, the integral evaluates to the same value. Hence, the last two terms of $\log Z(\boldsymbol{\theta}_{\text{MAP}})$ transform into $-\frac{d}{2}\log(2\pi) + \log\det\boldsymbol{H}(\varphi^{-1}(\boldsymbol{\psi}_{\text{MAP}}))$ in $\psi$-coordinates. This quantity is thus invariant under reparametrization since it behaves like standard functions.

### E.3.1 Experiment Setup

We use the toy regression dataset of size 150. Training inputs are sampled uniformly from $[0, 8]$, while training targets are obtained via $y = \sin x + \epsilon$, where $\epsilon \sim \mathcal{N}(0, 0.3^2)$. The network is a 1-hidden layer TanH network trained for 1000 epochs.

