\boldsymbol{\theta}^m \partial \boldsymbol{\theta}^n} - \boldsymbol{\Gamma}_{mn}^o \frac{\partial \mathcal{L}}{\partial \boldsymbol{\theta}^o} \right) \\
&= \frac{\partial \boldsymbol{\theta}^m}{\partial \boldsymbol{\psi}^i} \frac{\partial \boldsymbol{\theta}^n}{\partial \boldsymbol{\psi}^j} \boldsymbol{H}_{mn}.
\end{aligned}
$$

(8)

In the matrix form, we can write the above as $\tilde{\boldsymbol{H}} = \boldsymbol{J}^{-\top} \boldsymbol{H} \boldsymbol{J}^{-1}$, where $\boldsymbol{J}$ is the Jacobian of $\varphi$. Thus, the Riemannian Hessian at any $\boldsymbol{\theta}$ (not just at critical points) transforms just like the metric and thus invariant as discussed in Example 3. Note: this only holds when the term containing the connection coefficients $\boldsymbol{\Gamma}_{ij}^k$ is explicitly considered. In particular, the Euclidean Hessian does not follow this tensorial transformation under autodiff due to the fact that (i) $\boldsymbol{\Gamma}_{ij}^k = 0$ for any $i$, $j$, $k$ and thus dropped from the equation, *and* (ii) autodiff is not designed to handle advanced geometric objects like $\boldsymbol{\Gamma}_{ij}^k$.

## B.2 Hessian-Trace Under Reparametrization

Let $\mathcal{L} : \mathbb{R}^d \to \mathbb{R}$ be a function of $\mathbb{R}^d$ under the Cartesian coordinates and $\boldsymbol{G}$ a Riemannian metric. The **Riemannian trace** of the Hessian matrix $\boldsymbol{H}$ of $\mathcal{L}$ is defined by [47]:

$$(\mathrm{tr}_{\boldsymbol{G}} \boldsymbol{H})(\boldsymbol{\theta}) = \mathrm{tr}(\boldsymbol{G}(\boldsymbol{\theta})^{-1} \boldsymbol{H}(\boldsymbol{\theta})). \tag{9}$$

That is, it is defined as the standard trace of the Hessian operator $\boldsymbol{E}$.

Let $\varphi : \boldsymbol{\theta} \mapsto \boldsymbol{\psi}$ be a reparametrization on $\mathbb{R}^d$. Then, using (2) and the property $\mathrm{tr}(\boldsymbol{AB}) = \mathrm{tr}(\boldsymbol{BA})$ twice, the Riemannian trace of the Hessian transforms into

$$
\begin{aligned}
(\mathrm{tr}_{\tilde{\boldsymbol{G}}} \tilde{\boldsymbol{H}})(\boldsymbol{\psi}) &= \mathrm{tr}(\tilde{\boldsymbol{E}}(\boldsymbol{\psi})) \\
&= \mathrm{tr}((\boldsymbol{J}^{-1}(\boldsymbol{\psi}))^{-1} \boldsymbol{G}(\varphi^{-1}(\boldsymbol{\psi})) \boldsymbol{H}(\varphi^{-1}(\boldsymbol{\psi})) \boldsymbol{J}^{-1}(\boldsymbol{\psi})) \\
&= \mathrm{tr}(\boldsymbol{G}(\varphi^{-1}(\boldsymbol{\psi})) \boldsymbol{H}(\varphi^{-1}(\boldsymbol{\psi}))) \\
&= (\mathrm{tr}_{\boldsymbol{G}} \boldsymbol{H})(\varphi^{-1}(\boldsymbol{\psi})).
\end{aligned}
\tag{10}
$$

Since $\boldsymbol{\psi} = \varphi(\boldsymbol{\theta})$, we have that $(\mathrm{tr}_{\tilde{\boldsymbol{G}}} \tilde{\boldsymbol{H}})(\boldsymbol{\psi}) = (\mathrm{tr}_{\boldsymbol{G}} \boldsymbol{H})(\boldsymbol{\theta})$ for any given $\boldsymbol{\theta}$. Therefore, the trace of the Hessian operator (or the Riemannian trace of the Hessian) is invariant.

## B.3 Hessian-Eigenvalues Under Reparametrization

We use the setting from the preceding section. Recall that $\lambda$ is an eigenvalue of the linear map $\boldsymbol{E}(\boldsymbol{\theta}) = \boldsymbol{G}(\boldsymbol{\theta})^{-1} \boldsymbol{H}(\boldsymbol{\theta})$ on the tangent space at $z \in \mathbb{R}^d$ that is represented $\boldsymbol{\theta}$ if $\boldsymbol{E}(\boldsymbol{\theta}) \boldsymbol{v} = \lambda \boldsymbol{v}$ for an eigenvector $\boldsymbol{v} \in T_z \mathbb{R}^d$. We shall show that $\tilde{\lambda}$, the eigenvalue under under the reparametrization $\varphi : \boldsymbol{\theta} \mapsto \boldsymbol{\psi}$, equals the original eigenvalue $\lambda$.

Using the transformation rule of $\boldsymbol{E}(\boldsymbol{\theta})$ in (2) and the transformation rule of tangent vectors in (3), along with the relation $(\boldsymbol{J}^{-1}(\boldsymbol{\psi}))^{-1} = \boldsymbol{J}(\varphi^{-1}(\boldsymbol{\psi}))$, we get

$$
\begin{aligned}
\tilde{\boldsymbol{E}}(\boldsymbol{\psi}) \tilde{\boldsymbol{v}} &= \tilde{\lambda} \tilde{\boldsymbol{v}} \\

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

}})$ | Generalization gap [%] | Sharpness $\mathrm{tr}(\hat{\boldsymbol{H}}(\psi_{\mathrm{MAP}}))$ |
|---|---|---|---|
| ADAM | $\psi_{\mathrm{MAP}} = \boldsymbol{\theta}_{\mathrm{MAP}}$ | $7.2 \pm 0.2$ | $1929.8 \pm 61.2$ |
| SGD | $\psi_{\mathrm{MAP}} = \boldsymbol{\theta}_{\mathrm{MAP}}$ 
 $\psi_{\mathrm{MAP}} = \frac{1}{2}\boldsymbol{\theta}_{\mathrm{MAP}}$ 
 $\psi_{\mathrm{MAP}} = 2\boldsymbol{\theta}_{\mathrm{MAP}}$ | $5.2 \pm 0.2$ | $1531.8 \pm 14.2$ 
 $6143.7 \pm 60.8$ 
 $383.6 \pm 3.3$ |

Since the integral is the normalization constant of the Gaussian $\mathcal{N}(\boldsymbol{\theta} \mid \boldsymbol{\theta}_{\mathrm{MAP}}, \boldsymbol{H}(\boldsymbol{\theta}_{\mathrm{MAP}}))$, we obtain the Laplace log-marginal likelihood (LML):

$$\log Z(\boldsymbol{\theta}_{\mathrm{MAP}}) = -\mathcal{L}_{\mathrm{MAP}}(\boldsymbol{\theta}_{\mathrm{MAP}}) - \frac{d}{2}\log(2\pi) + \log \det \boldsymbol{H}(\boldsymbol{\theta}_{\mathrm{MAP}}).$$

Notice that $\boldsymbol{H}(\boldsymbol{\theta})$ is a bilinear form, acting on the tangent vector $\boldsymbol{d}(\boldsymbol{\theta}_{\mathrm{MAP}}) := (\boldsymbol{\theta} - \boldsymbol{\theta}_{\mathrm{MAP}})$. Under a reparametrization $\varphi : \boldsymbol{\theta} \mapsto \psi$ with $\psi_{\mathrm{MAP}} = \varphi(\boldsymbol{\theta}_{\mathrm{MAP}})$, the term inside the exponent in (15) transforms into

$$-\frac{1}{2}\big(\boldsymbol{J}(\varphi^{-1}(\psi_{\mathrm{MAP}}))\boldsymbol{d}(\varphi^{-1}(\psi_{\mathrm{MAP}}))\big)^{\top}\big(\boldsymbol{J}^{-1}(\psi_{\mathrm{MAP}})^{\top}\boldsymbol{H}(\varphi^{-1}(\psi_{\mathrm{MAP}}))\boldsymbol{J}^{-1}(\psi_{\mathrm{MAP}})\big)$$
$$\boldsymbol{J}(\varphi^{-1}(\psi_{\mathrm{MAP}}))\boldsymbol{d}(\varphi^{-1}(\psi_{\mathrm{MAP}})),$$

due to the transformations of the tangent vector and the bilinear-Hessian. This simplifies into

$$\exp\left(-\frac{1}{2}\boldsymbol{d}(\varphi^{-1}(\psi_{\mathrm{MAP}}))^{\top}\boldsymbol{H}(\varphi^{-1}(\psi_{\mathrm{MAP}}))\boldsymbol{d}(\varphi^{-1}(\psi_{\mathrm{MAP}}))\right)$$

which always equals the original integrand in (15). Thus, the integral evaluates to the same value. Hence, the last two terms of $\log Z(\boldsymbol{\theta}_{\mathrm{MAP}})$ transform into $-\frac{d}{2}\log(2\pi) + \log \det \boldsymbol{H}(\varphi^{-1}(\psi_{\mathrm{MAP}}))$ in $\psi$-coordinates. This quantity is thus invariant under reparametrization since it behaves like standard functions.

### E.3.1 Experiment Setup

We use the toy regression dataset of size 150. Training inputs are sampled uniformly from $[0, 8]$, while training targets are obtained via $y = \sin x + \epsilon$, where $\epsilon \sim \mathcal{N}(0, 0.3^2)$. The network is a 1-hidden layer TanH network trained for 1000 epochs.