# OpenReview forum: "The Geometry of Neural Nets' Parameter Spaces Under Reparametrization"
_NeurIPS.cc/2023/Conference — NeurIPS 2023 spotlight_

### Official Review · Reviewer_togv · 2023-06-29

**Soundness:** 3 good
**Presentation:** 3 good
**Contribution:** 3 good
**Rating:** 7
**Confidence:** 4

**Summary:**

The paper discusses reparametrizations of parameter spaces and the implied transformation rules for quantities like gradients, Hessians or probability densities.
Parameter spaces are interpreted as Riemannian manifolds $M=\mathbb{R}^d$ and the quantities of interest are coordinate independent geometric objects like tangent vectors, covectors, tensors, or volume forms.
The transformation laws of such objects are well known in differential geometry,
however, as the authors argue, they are in deep learning applications often disregarded.
In particular, the metric tensor on $M=\mathbb{R}^d$ is often taken to be $G=\mathbb{I}$ and therefore dropped from the equations.
Its transformation when changing coordinates is then naively forgotten, that is, one uses again metric coefficients $\hat{G}=\mathbb{I}$ in the new coordinates, which correspond geometrically to a _different Riemannian geometry_, such that the algorithms become coordinate dependent.
The contribution of the paper is to point out these disregarded transformation rules and to discuss how the quantities should actually transform.

Section 2 introduces the mathematical setting and explains transformation rules of geometric quantities, giving in particular examples of how their transformation laws are consistent by canceling out in different tensor contractions.
The third section discusses more specific quantities of interest in deep learning.
Firstly, it considers the Hessian matrix determinant as measure of flatness of parameter landscapes.
The authors argue that this measure is not well-suited since it depends on the choice of coordinates.
They propose to use the determinant of $G^{-1}H$ instead, which is invariant under reparametrizations.
Secondly, it considers loss gradients $\mathrm{grad}\mathcal{L} := G^{-1}\nabla\mathcal{L}$.
On $M=\mathbb{R}^d$, the trivial metric $G=\mathbb{I}$ is usually dropped, which leads again to inconsistent transformations of the gradient.
Lastly, Section 3.3 discusses probability density functions (pdfs).
When being expressed relative to the Lebesgue measure on $M=\mathbb{R}^d$, the pdfs transform with the well known Jacobian determinant factor.
However, as the density may be stretched out or condensed in this procedure, the densities' mode may not be preserved.
It is hence more suitable to express the density relative to the Riemannian volume form: as this form transforms itself already with the Jacobian determinant factor, the density relative to it remains invariant, which preserves in particular its modes.
After discussing related work in section 4, the fifth section considers some applications, arguing in particular
that NTK and standard parametrizations or neural networks are not just reparametrizations, but geometrically truly different models (section 5.1) and
that the Laplace marginal likelihood is invariant under reparametrization (section 5.2).
It investigates furthermore the effect of using the Hessian or $G^{-1}H$ in (preconditioned) optimizers like ADAM.

**Strengths:**

The authors observe that many of the mathematical formulations used in deep learning are from a geometric perspective not coordinate independent, which is a fundamental feature that any consistent mathematical theory should satisfy.
Its main contribution lies in pointing out this issue and discussing how the equations can be fixed.
While this contribution is not "original" in the sense that it would be a novel idea, making researchers aware of it is of utmost significance since a coordinate independent formulation of algorithms is a fundamental requirement.
It is hard to judge how clear the paper is for someone without background in differential geometry, but it is kept simple and is certainly easy to reed for someone knowing about differential geometry.

**Weaknesses:**

A main weakness of the paper is that the mathematical formulation could be more precise at some points.
I would usually not be too strict in deep learning, however, as the paper sets out to fix the inconsistent use of mathematics, it should be more precise.
These weaknesses should be easy to fix - more details follow in the next paragraphs.

Firstly, the considered coordinate charts and hence reparametrizations are _global_ homeomorphisms.
This is in principle possible, but excludes practically relevant choices like polar or spherical coordinates, which are not global homeomorphisms.
Polar coordinates are, in fact, used as an example right after saying that charts should be global homeomorphisms.
The only clean way around this issue (and to include polar coordinates) would be to admit the usual atlases of _local_ charts and study reparametrizations as usual on the intersections of charts.
Specifically, there should be charts
$\theta: U^\theta \to \theta(U^\theta) \subseteq \mathbb{R}^d$ and $\psi: U^\psi \to \psi(U^\psi) \subseteq \mathbb{R}^d$
on domains $U^\theta\subseteq M=\mathbb{R}^d$ and $U^\psi\subseteq M=\mathbb{R}^d$
with transition maps $\varphi: \theta(U^\theta\cap U^\psi) \to \psi(U^\theta\cap U^\psi)$.
Note that the main results of the paper will still hold in this more general setting.

Secondly, the paper keeps talking about a dubious concept of "_equivariance under reparametrization_", which is non-standard and, while looking somewhat similar to the usual concept of group equivariance, is different from it and more confusing than it is enlightening.
It is introduced at the end of the second section,
assuming parameter space reparametrizations $\varphi:\Theta\to\Psi$
and functions which simultaneously seem to satisfy $F: \Theta\to\Theta$ and $F: \Psi\to\Psi$.
This would in principle require $\Theta=\Psi$, while actually only $\Theta\cong\mathbb{R}^d\cong\Psi$ is demanded initially.
Furthermore, this does not work in the clean formulation with local charts suggested above, since then
$\varphi: \theta(U^\theta\cap U^\psi) \to \psi(U^\theta\cap U^\psi)$
and the upper and lower arrows would be
$F^\theta: \theta(U^\theta\cap U^\psi) \to \theta(U^\theta\cap U^\psi)$ and
$F^\psi: \psi(U^\theta\cap U^\psi) \to \psi(U^\theta\cap U^\psi)$, respectively.
The corresponding diagram would be the usual coordinate independence transformation rule $F^\psi = \varphi\circ F^\theta\circ \varphi^{-1}$, which is _not_ an equivariance condition.
It can in general also not be made to one by setting $F^\psi = F^\theta$, as this would require the equality of $\theta(U^\theta\cap U^\psi) = \psi(U^\theta\cap U^\psi)$ in the first place.

The concept of equivariance is subsequently used in a rather unspecific way in the paper.
Section 3.2 talks about and is titled "equivariance of gradient descent", however, this is not made precise, i.e. no equation is mentioned which follows the commutative diagram in the author's definition of equivariance.
Furthermore, equivariance is a property of _functions_, and it is not entirely clear to me how one should interpret the gradient descent algorithm as such.
Section 3.3 is titled "equivariance of probability densities", which seems to refer to equation 4, i.e. $q_\Psi^G(\psi) = q_\Theta^G(\varphi^{-1}(\psi))$.
However, this is just a coordinate independence equation, since the functions $q_\Psi^G$ and $q_\Theta^G$ on the left and right hand side differ from each other (they are not the same $F$ as in the diagram).

The issue of this dubious notion of "equivariance" is easily fixed by removing it from the paper
and referring to it as usual as coordinate independence or covariance.

**Questions:**

The issue of a coordinate independent formulation of deep learning algorithms was previously studied in the publication "Equivariant and coordinate independent convolutional networks" by Weiler et al. (2021), which should be mentioned.
I would furthermore be very interested in how their coordinate independence of feature vectors and neural network operations relates to the coordinate independence of parameter spaces in the current submission?

Further minor suggestions and corrections beyond what I wrote above follow. These should be very easy to fix.

Line 50 states that
_"Intrinsic ... means that objects ... must be independent of ... coordinate system"_.
This concept is called coordinate independence or covariance.
"Intrinsic" refers instead to "not extrinsic", where extrinsic properties are geometric properties depending on an embedding of the manifold in some ambient space (e.g. sectional curvatures).

Charts are defined as homeomorphisms, but the paper considers smooth manifolds.
The smooth structure is actually only respected (and defined via) smooth charts, i.e. diffeomorphisms.

It would probably be helpful to note that the "standard choice" of global chart in line 97 is the canonical identity map $\mathrm{id}_{\mathbb{R}^d}$.

That the choice of coordinate system is not unique is somewhat tricky,
as there exists the _canonical_ global chart $\mathrm{id}_{\mathbb{R}^d}$ mentioned by the authors.
If the manifold was just taken as Euclidean space, i.e. with metric but without the canonical coordinates of $\mathbb{R}^d$,
global charts would still not be arbitrary, but one could restrict to isometries, which are defined up to transition maps in the Euclidean group $\mathrm{E}(d)$.
This raises the question why we are considering general diffeomorphisms in the first place?
Is the metric just an arbitrary choice (which should of course still be respected, requiring coordinate independence)?
It would be great if the authors could discuss this point.

Line 120 mentions that
"Under a coordinate system, one can think of both tangent vectors and covectors as vectors in the sense of linear algebra, i.e., tuples of numbers",
however, also abstract coordinate free vectors are part of linear algebra.
I would just write that vectors and covectors are in coordinates represented by numerical coefficient vectors, i.e. tuples of numbers.

Line 123: The metric is not only positive definite, but also symmetric.

I am confused about the "surjective everywhere" in line 134.
Is the problem with bijectivity not that the mapping from parameters to models is in general non-injective?

It might be helpful to mention somewhere around line 193 that the $\Gamma_{ij}^k$ are called Christoffel symbols. This way the reader unfamiliar with these concepts can read up on it.

**Limitations:**

As the paper does not propose a new method but comments on the mathematical formulation of theories, limitations do not really apply.

---

> ### Author Rebuttal · Authors · 2023-08-07
>
> Thanks a lot for your extensive review! Your write-up on the summary and strengths of our paper is completely spot on!
>
> Here we will address your major comments and questions. Minor comments and suggestions will be implemented directly in the text. See also our ["global" response](https://openreview.net/forum?id=vtLNwa6uX0&noteId=CvzNeGzNp7) for general discussion.
>
> **Precise mathematical formulation** You are right that it can indeed be more precise. Your suggestions on the usage of local-chart formulation and removing the current notion of “equivariance” & changing it into “coordinate-independence” are spot on and we will implement them in the paper. On the other hand, we are trying to make the text accessible to a broader audience in deep learning—this is our target community. Note that other reviewers, who seem to fall into this category, mentioned that our paper is accessible (**_R.ecDa_**, **_R.Poha_**) and insightful (**_R.LwSR_**). So, we will follow your suggestion by showing predominantly intuition and figures in the main text and deferring the extra mathematical details to the appendix.
>
> **Weiler et al. (2021)** They tackle the problem of “geometric deep learning”, where the manifold of interest is the input (and feature) space. So, at a high level, our work—which focuses on the parameter space—is complementary to theirs. E.g., one benefits from our work when measuring the sharpness of the loss landscape of a gauge-equivariant NN. While indeed their theory tells us how to transform parameters (i.e. convolution kernels in their case) under a gauge transformation (a group action), they are compatible with our work in the same way that the symmetry of a manifold is compatible with coordinate independence of the same manifold.
>
> **Choice of coordinate systems, is the metric arbitrary?** $\mathbb{R}^n$ with the canonical global coordinates is often the default choice for the parameter space of deep networks and only the metric is varied (e.g. in natural gradient methods and normalizing flows—the latter can be seen as a metric-learning mechanism through a non-invariant coordinate transformation acting on the canonical coordinates). Even for geometric-focused NNs like gauge equivariant nets [1], where strong manifold assumptions are imposed in the input/feature space, to our knowledge no further manifold assumption is applied on the parameter space, other than possibly the metric (e.g. using SGD vs. ADAM during the optimization). In this sense, considering general diffeomorphisms is useful since our work then provides the coordinate-independence guarantee and preservation of the metric in broad deep-learning applications.
>
> **Surjective everywhere** The problem is that the map $\theta \mapsto f(X; \theta)$ is almost always a submersion for an overparametrized network [2, 3], and certainly not a diffeomorphism. Meanwhile, the requirement for pulling back a metric is that the map must be an immersion. (Note that (local) diffeomorphism implies immersion.) We will rephrase “surjective everywhere” into “non-injective everywhere” to make this point clearer.
>
> If you have further suggestions, we always welcome them! Thanks again for your great comments and suggestions!
>
> **References**
>
> [1] Cohen, Taco, et al. "Gauge equivariant convolutional networks and the icosahedral CNN." ICML 2019.
>
> [2] Zhang, Guodong, James Martens, and Roger B. Grosse. "Fast convergence of natural gradient descent for over-parameterized neural networks." NeurIPS 2019.
>
> [3] Karakida, Ryo, and Kazuki Osawa. "Understanding approximate Fisher information for fast convergence of natural gradient descent in wide neural networks." NeurIPS 2020.

---

> > ### Comment · Reviewer_togv · 2023-08-12
> >
> > The authors addressed the issues raised in "weaknesses" by promising to rewrite the paper accordingly. I don't have any remaining questions.
> >
> > Please make sure to include a discussion of the relation to Cohen et al. and Weiler et al. in the paper. Note that they also do not require symmetries of the manifold itself, but just consider "gauge symmetries" in the parametrization/coordinates of tangent spaces. Equivariance under symmetries of the manifold may be induced by this coordinate independence.

---

### Official Review · Reviewer_LwSr · 2023-07-05

**Soundness:** 4 excellent
**Presentation:** 4 excellent
**Contribution:** 4 excellent
**Rating:** 8
**Confidence:** 3

**Summary:**

The paper shows that reparameterizations of neural networks can be understood uisng Riemannian geometry. They first show how a reparameterization of a neural network's parameters can be expressed via a Riemannian metric which then yields transformation rules that can be applied to any function on the parameters. The paper uses this as basis to show why the determinant, trace, and Eigenvalues of the loss Hessian are not invariant under reparameterization, and how applying the correct transformation rules yields reparameterization invariance. The same is shown for gradient descent and probability densities. Lastly, the paper applies this to infinite-width neural networks (showing that the NTK is not a reparameterization of a standard infinite-width Bayesian NN), to the Laplace marginal likelihood, and to preconditioned optimizers.

**Strengths:**

- Viewing reparameterizations from the perspective of Riemannian geometry brings much needed clarity to the discussion.
- The paper is excellently written and insightful, a pleasure to read.
- The paper discusses a wide range of implications relevant to machine learning.

**Weaknesses:**

- The discussion on flatness-based generalization measures does not address existing generalization bounds for reparameterization-invariant flatness measures (see questions).

**Questions:**

- It might be interesting to see how a very simple reparameterization (e.g., multiplying one layer with a constant c and the next with 1/c for ReLU NNs) can be interpreted in Riemannian geometry. I.e., what is G in that case, how would the transformed Hessian look like? While this might not fit into the main text, it would make for a great practical example in the appendix for readers (like myself) not too familiar with Riemannian geometry.
- The relation between flatness and generalization in light of reparameterizations has been established theoretically in [1].
- Is it possible to interpret relative flatness [1] as an invariant transformation? That is, is it possible that $G^-1(\theta) = ||\theta||^2_2$?

[1] Petzka, Henning, et al. "Relative flatness and generalization." Advances in neural information processing systems 34 (2021): 18420-18432.

**Limitations:**

The paper is clear about the assumptions and limitations.

---

> ### Author Rebuttal · Authors · 2023-08-07
>
> Thanks a lot for your positive review! Please see also our ["global" response](https://openreview.net/forum?id=vtLNwa6uX0&noteId=CvzNeGzNp7) for a general discussion. Here, we address your major comments. All other comments and suggestions are implemented directly in the paper.
>
> You are completely right that our work is positioned in such a way as to bring much-needed clarity to the behavior of neural networks’ parameter spaces under reparametrization. We are glad to hear that our work can be followed easily and gives insights to you, as a researcher outside of Riemannian geometry.
>
> **Relative flatness of Petzka et al.** We note that Petzka et al., while using the term “reparametrization” in their paper, actually tackle the “symmetry” problem discussed in Sec. 1 in our paper. Reparametrization is a specific transformation of the parameter space under a smooth invertible map (diffeomorphism)—in the language of calculus, it is essentially the change-of-variable formula, e.g. in integration by substitution. Symmetry, meanwhile is defined through group actions [1], e.g. under rescaling of the weights where not just one scaling factor $c > 0$ is considered, but all of $c \in \mathbb{R}\_{>0}$ (the space $\mathbb{R}_{>0}$ here is the group).
>
> In any case, both kinds of invariance (under symmetry and reparametrization) are important as we mentioned in our paper (see also [2], Sec. 3 and Sec. 5) since in general one does not imply the other. (See our "global" response.) They thus complement each other and our work complements Petzka et al.’s relative flatness which provides an invariant measure for generalization under rescaling group actions, but is not invariant under reparametrization. (Notice that relative flatness is defined through the non-invariant Hessian trace, and we show in our paper how to make it invariant.) In other words, our work makes relative flatness invariant to _both_ rescaling group actions and reparametrization.
>
> **Interpretation of as an invariant transformation** The symmetry $f(c \theta) = f(\theta)$ can be written as a group action of the group $\mathcal{G} := \mathbb{R}_{>0}$ on the manifold $\Theta := \mathbb{R}^n$ by multiplication. Your intuition is spot on that $\mathcal{G}$-invariant quantities such as relative flatness can be seen as quantities in the "symmetry-free space". In the above case, one can think of relative flatness as a generalization metric on the quotient space $\Theta / \mathcal{G}$, which happens to be the sphere $\mathbb{S}^{n-1} = \\{ \theta \in \Theta : \Vert \theta \Vert^2_2 = 1 \\}$.
>
> **Example with simple reparametrization** As we have discussed before, the rescaling "reparametrization" is better described as "symmetry" and thus is not suited as an example in this paper. But, we are happy to give a simple step-by-step example of reparametrization and its effect on the metric, Hessian, etc. in the appendix. We will do so by expanding Example 1a, i.e., using the transformation $\theta = \log \psi$.
>
> [Our answer to **_R.ecDa_**](https://openreview.net/forum?id=vtLNwa6uX0&noteId=1K4jej38ao) might also be of interest to you.
>
> Thanks again and please let us know if you have further comments!
>
> **References**
>
> [1] Kunin, Daniel, et al. "Neural Mechanics: Symmetry and Broken Conservation Laws in Deep Learning Dynamics." ICLR, 2020
>
> [2] Dinh, Laurent, et al. "Sharp minima can generalize for deep nets." ICML, 2017.

---

> > ### Comment · Reviewer_LwSr · 2023-08-14
> > **Answer to authors**
> >
> > I thank the authors for their reply. Both the reply and the other reviews keep me convinced that this is a good paper. I keep my score.

---

### Official Review · Reviewer_Poha · 2023-07-07

**Soundness:** 3 good
**Presentation:** 3 good
**Contribution:** 3 good
**Rating:** 6
**Confidence:** 1

**Summary:**

This work analyzes the invariances and non-invariances of model reparameterization in machine learning. The authors show that, if we account for Riemannian metrics in parameter spaces, then many quantities thought to be not invariant are in fact invariant to reparameterization. Thus, by properly applying transformation rules on geometric quantities, we can obtain equivariant or invariant functions on parameter space.


**Strengths:**

1. Covers applications of this type of thinking in several areas of machine learning.
2. Good, careful exposition of geometric concepts and calculations.
3. The note on the utility of non-invariant reparameterization for normalizing flows and optimization is interesting.
4. Overall, this work gives a useful perspective that helps analyze ML models (e.g. Section 5.2), and will hopefully give actionable insights to improve them (e.g. other metrics instead of Fisher).


**Weaknesses:**

1. I am not sure about the utility of the suggested method of measuring sharpness, and I would appreciate if the authors could comment on this. Indeed, the sharpness of ReLU networks depend on the scale of the weights chosen. However, [Du et al. 2018] shows that there is some implicit bias so that arbitrary scales of the weights are not converged to by GD in practice. Also, there are generalization results in terms of (I believe) non-invariant Hessian trace [Ding et al. 2023].

[Du et al. 2018] Algorithmic Regularization in Learning Deep Homogeneous Models: Layers are Automatically Balanced. NeurIPS 2018
[Ding et al. 2023] Flat minima generalize for low-rank matrix recovery. arXiv 2023.


**Questions:**

n/a

**Limitations:**

Discussion of limitations on Page 2.

---

> ### Author Rebuttal · Authors · 2023-08-07
>
> Thank you for your positive review! Please see our ["global" response](https://openreview.net/forum?id=vtLNwa6uX0&noteId=CvzNeGzNp7) for a general discussion. Here, we address your specific comments.
>
> **Du et al.** They focus on the invariance of ReLU networks under the scaling symmetry, while we are tackling the problem of invariance under reparametrization. This difference is also made clear by Dinh et al. 2015, Sec. 3 (symmetry) vs. Sec. 5 (reparametrization). See also [our answers to **_R.ecDA_**](https://openreview.net/forum?id=vtLNwa6uX0&noteId=1K4jej38ao) and our "global" response.
>
> **Ding et al.** Indeed they showed generalization results with the non-invariant Hessian trace. However, their results will have the pathologies that we discussed in our paper and in Sec. 5 of Dinh et al. Our work is compatible with them in that it provides a guardrail for their work: we give the necessary steps to make their results resistant to pathologies under reparametrization.
>
> In any case, we added both works to the related work section of our paper. Please let us know if you have further comments or questions!
>
> **References**
>
> [1] Dinh, Laurent, et al. "Sharp minima can generalize for deep nets." ICML, 2017.

---

> > ### Comment · Reviewer_Poha · 2023-08-17
> >
> > We thank the author for their rebuttal. I think certain readers would appreciate the addition of discussion for these two papers. I have no further questions.

---

### Official Review · Reviewer_ecDa · 2023-07-07

**Soundness:** 3 good
**Presentation:** 2 fair
**Contribution:** 3 good
**Rating:** 6
**Confidence:** 3

**Summary:**

Under model reparametrization, Hessian-based flatness measures, optimization trajectories, and probability densities are not invariant. Motivate by these inconsistencies, this paper studies the invariance associated with the reparametrization of neural networks. By viewing the parameter space as a Riemannian manifold, the authors show that the invariance and equivariance under reparametrization is preserved by explicitly including the metric when computing geometric objects such as the Hessian. The authors point out that acknowledging the metric helps in measuring the flatness of minima, optimization, and probability-density maximization.


**Strengths:**

This paper draws attention to the nature of reparametrization through Riemannian geometry. By introducing a framework that transforms representations of geometric objects to keep them invariant under reparametrizations, the paper provides a useful tool in comparing properties of neural networks after reparametrization.

I appreciate the authors’ effort to make the derivations both mathematically rigorous and accessible. In particular, since the parameter space is usually Euclidean, it makes sense to present most of the material in linear algebra terminologies instead of the more general Riemannian geometry.

**Weaknesses:**

The novelty of this paper seems limited. As the authors also mention, the lack of invariance under reparametrization has been observed before. The transformation of various quantities in reparametrization has also been discussed (see below). The discoveries on the application side are also not well-presented. As a result, it is not clear what the main contributions are.

Some parts of the paper could be explained in more details. (a) The goal of section 5.1 is not clear. Is the goal to show that SP and NPT are different because they cannot be seen as reparametrizations of each other? (b) The significance of section 5.2 might be clearer if the authors could add a sentence to give a general intuition for Equation 5.

The transformation of the Hessian and gradient under reparametrization has been discussed in a previous paper that is not cited [1]. Could the authors comment on how their approach in section 3 is different?

[1] Kunin, Daniel, et al. "Neural Mechanics: Symmetry and Broken Conservation Laws in Deep Learning Dynamics." International Conference on Learning Representations. 2020. (Appendix A)


**Questions:**

- By comparing the definitions on page 1, it seems that invariance under symmetry is a special case of invariance under reparametrization. Can the reparametrization studied in this paper be viewed as more general than previous works on weight-space symmetries?
- Could the authors elaborate on the intuition on why we should explicitly include the metric when comparing the sharpness of the solution found by different optimizers, as suggested in section 5.3?

**Limitations:**

The authors state limitations at the end of the introduction section. There are no potential negative societal impacts of the work.

---

> ### Author Rebuttal · Authors · 2023-08-07
>
> Thank you very much for your feedback! Here we address your major comments/questions. We incorporated your suggestions into the text directly. Please see also our ["global" response](https://openreview.net/forum?id=vtLNwa6uX0&noteId=CvzNeGzNp7) for a general discussion.
>
> **Novelty** The main contribution of our present work is to show that invariance under reparametrization of many quantities relevant to neural nets is natural when considering the correct transformation of geometric objects. While indeed previous work has discussed non-invariance under reparametrization, they either accept it as a fact (e.g. Sec. 5 of [1]) or come up with a special metric/method to overcome this issue (e.g. [2]). Our argument, meanwhile, is broadly applicable since no assumption about the metric is imposed. Our work thus provides a guardrail against past & future confusions regarding invariance. We hope to invoke more interest in this topic by providing some example applications in Sec. 5. Finally, please note that other reviewers mentioned that our work provides a new perspective (**_R.Poha_**), brings much-needed clarity (**_R.LwSr_**), and is useful since making ML researchers aware of the discussed topic is of utmost significance (**_R. togv_**).
>
> **Sec. 5.1** Our goal is to clear up confusion about the term “parametrization” in SP and NTP. In the preceding sections we have argued that if two parametrizations are connected by a diffeomorphism, they represent a single function. But since SP and NTP, despite their monikers, are not reparametrization of each other, then they represent two different functions. This clears up the confusion why they have different limiting behavior (in terms of the NTK and NNGP kernels) and calls for a more suitable way of analyzing them.
>
> **Reparametrization vs symmetry** When one fixes the bijective transformation $T$, and it happens to be smooth, then indeed they can be seen as the same, e.g. in the normalized NN example, the scaling $c$ is fixed. However, in general $T$ is not fixed, i.e. we consider instead a _family_ of invertible $\Theta \to \Theta$ and so they are inherently different. Invariance under symmetries is better expressed in terms of group actions from (Lie) group theory, i.e. the map $T$ should instead be defined as $T: \mathcal{G} \times \Theta \to \Theta$ where $\mathcal{G}$ is a group. For normalized NNs, $\mathcal{G} = \mathbb{R}_{>0}$, i.e. we take into account _all_ scaling factors $c > 0$ in $\mathcal{G}$ instead of fixing it. The definition of symmetry on page 1 is indeed incomplete (missing the group term) and we have updated it. Please note that this missing term does not affect the discussion since we focus on reparametrization, not symmetry. Note also that the group-action definition is standard in the literature, including in Kunin et al.
>
> **Kunin et al.** They analyze the gradient and the Hessian under symmetry, not reparametrization (see above). Their work therefore tackles a different problem than ours. In particular they study invariance under group actions, while we study invariance under diffeomorphism. Both are important (as written in our paper and in e.g. [1] Sec. 3 and Sec. 5) and our work is complementary to theirs—our geometric insights further enhance their symmetry-invariant gradient and Hessian formulations with reparametrization invariance. We added their work to the citation list. Thanks for pointing out their work!
>
> **Why include the metric in sharpness** There are at least two reasons why. _First_, as we have shown in Sec. 2.2.1 and Sec. 3.1, this is the geometrically principled way to compute Hessian-based sharpness (trace, det, eigenvalues), and this naturally yields invariance under reparametrization and solves the problem shown by [1, Sec. 5]. _Second_, when using a preconditioned gradient descent (PGD), the metric-infused sharpness yields similar behaviors as standard gradient descent [3], allowing for direct comparisons. Indeed, PGD is essentially just a GD acting on a space with different geometry induced by the metric. Moreover, by taking into account the parameter-space metric, this reveals that the loss landscape geometry under ADAM’s metric is actually much sharper than that assumed in GD (our Fig. 5)---this information might be useful for future work.
>
> [Our answer to **_R.LwSr_**](https://openreview.net/forum?id=vtLNwa6uX0&noteId=OHnkKsN22C) regarding symmetry vs reparametrization might also interest you.
>
> Please let us know if you have further comments/questions/suggestions!
>
> **References**
>
> [1] Dinh, Laurent, et al. "Sharp minima can generalize for deep nets." ICML, 2017.
>
> [2] Jang, Cheongjae, et al. "A reparametrization-invariant sharpness measure based on information geometry." NeurIPS, 2022.
>
> [3] Cohen, Jeremy M., et al. "Adaptive gradient methods at the edge of stability." arXiv preprint arXiv:2207.14484 (2022).

---

> > ### Comment · Reviewer_ecDa · 2023-08-15
> >
> > Thank you for the response. I now have a better understanding of the significance of invariance under reparametrization. I also appreciate the clarification on the difference between invariance under reparametrization and symmetry. I have increased my score accordingly.

---

### Author Rebuttal · Authors · 2023-08-07

**To all reviewers:** Thank you very much for your input! To supplement the responses to your individual reviews, here we would like to address the common questions and comments.

Our work focuses on addressing _invariance under reparametrization_, i.e. under change of variable from the point of view of calculus or the concept of coordinate independence from the point of view of differential geometry. In particular, it studies the invariance of quantities like gradients and Hessians when the coordinates of the parameter space are mapped into _new_ coordinates. This is different from _invariance under symmetry_, where the invariance is studied under a _group_ acting on a _fixed_ choice of coordinates of the parameter space.

Crucially, these two concepts are compatible with each other. Indeed, from the differential geometry point of view, coordinate independence is an _inherent_ property of a manifold as **_R.togv_** also pointed out; symmetry is a property that _can be_ studied further on that manifold—every manifold is coordinate-independent, but not every manifold has symmetry. Thus, our work complements previous works and provides a guardrail for future works that focus on studying invariance under symmetry.  For example, our work enhances the works of Kunin et al. [1], Petzka et al. [2], and Cohen et al. [3, 4], where they addressed invariances under various symmetries but are still susceptible to pathologies under reparametrization. (See our individual responses for more details.) Please note also that even though these previous works often used the term "reparametrization", their problem is better termed as "symmetry", as per the differential-geometric definitions we use.

**References**

[1] Kunin, Daniel, et al. "Neural Mechanics: Symmetry and Broken Conservation Laws in Deep Learning Dynamics." ICLR, 2020

[2] Petzka, Henning, et al. "Relative flatness and generalization." NeurIPS, 2021.

[3] Cohen, Taco, et al. "Gauge equivariant convolutional networks and the icosahedral CNN." ICML 2019.

[4] Weiler, Maurice, et al. "Coordinate Independent Convolutional Networks--Isometry and Gauge Equivariant Convolutions on Riemannian Manifolds." arXiv preprint arXiv:2106.06020 (2021).

---

### Decision · Program_Chairs · 2023-09-21

**Decision:**

Accept (spotlight)

**Comment:**

The reviewers reached unanimous agreement that the paper merits acceptance. They noted that the paper gives a rigorous yet accessible treatment of reparameterization of neural networks from a differential geometry perspective, providing much needed clarity to the topic. Reviewers also appreciated the fact that the paper discusses a wide range of implications for ML. As such I recommend accepting the paper as a spotlight presentation.